# LncRNA *Snhg3* aggravates hepatic steatosis via PPARγ signaling

**Xianghong Xie[1], Mingyue Gao[1], Wei Zhao[1], Chunmei Li[1], Weihong Zhang[2], Jiahui Yang[2], Yinliang Zhang[1], Enhui Chen[3], Yanfang Guo[1], Zeyu Guo[1], Minglong Zhang[3], Ebenezeri Erasto Ngowi[4,5], Heping Wang[1], Xiaoman Wang[1], Yinghan Zhu[3], Yiting Wang[1], Xiaolu Li[1], Hong Yao[2], Li Yan[3], Fude Fang[1], Meixia Li[6]\*, Aijun Qiao[4,5]\*, Xiaojun Liu[1]\***

[1]Department of Biochemistry & Molecular Biology, State Key Laboratory of Common Mechanism Research for Major Diseases, Institute of Basic Medical Sciences Chinese Academy of Medical Sciences & School of Basic Medicine Peking Union Medical College, Beijing, China; [2]Department of Microbiology and Immunology, Shanxi Medical University, Taiyuan, China; [3]Department of Pathophysiology, Institute of Basic Medical Sciences Chinese Academy of Medical Sciences & School of Basic Medicine Peking Union Medical College, Beijing, China; [4]Shanghai Institute of Materia Medica, Chinese Academy of Sciences, Shanghai, China; [5]Zhongshan Institute for Drug Discovery, Shanghai Institute of Materia Medica, Chinese Academy of Sciences, Zhongshan, China; [6]State Key Laboratory of Brain and Cognitive Science, Institute of Biophysics, Chinese Academy of Sciences, Beijing, China

**\*For correspondence:**
limeixia@ibp.ac.cn (ML);
qiaoaijun@simm.ac.cn (AQ);
xiaojunliu@ibms.pumc.edu.cn
(XL)

**Competing interest:** The authors declare that no competing interests exist.

**Abstract** LncRNAs are involved in modulating the individual risk and the severity of progression in metabolic dysfunction-associated fatty liver disease (MASLD), but their precise roles remain largely unknown. This study aimed to investigate the role of lncRNA *Snhg3* in the development and progression of MASLD, along with the underlying mechanisms. The result showed that *Snhg3* was significantly downregulated in the liver of high-fat diet-induced obesity (DIO) mice. Notably, palmitic acid promoted the expression of *Snhg3* and overexpression of *Snhg3* increased lipid accumulation in primary hepatocytes. Furthermore, hepatocyte-specific *Snhg3* deficiency decreased body and liver weight, alleviated hepatic steatosis and promoted hepatic fatty acid metabolism in DIO mice, whereas overexpression induced the opposite effect. Mechanistically, *Snhg3* promoted the expression, stability and nuclear localization of SND1 protein via interacting with SND1, thereby inducing K63-linked ubiquitination modification of SND1. Moreover, *Snhg3* decreased the H3K27me3 level and induced SND1-mediated chromatin loose remodeling, thus reducing H3K27me3 enrichment at the *Pparg* promoter and enhancing PPARγ expression. The administration of PPARγ antagonist T0070907 improved *Snhg3*-aggravated hepatic steatosis. Our study revealed a new signaling pathway, *Snhg3*/SND1/H3K27me3/PPARγ, responsible for mice MASLD and indicates that lncRNA-mediated epigenetic modification has a crucial role in the pathology of MASLD.

## eLife assessment

This study provides **useful** evidence substantiating a role for long noncoding RNAs in liver metabolism and organismal physiology. Using murine knockout and knock-in models, the authors invoke a previously unidentified role for the lncRNA Snhg3 in fatty liver. The revised manuscript has improved and most studies are backed by **solid** evidence but the study was found to be **incomplete** and will require future studies to substantiate some of the claims.

## Introduction

Non-alcohol fatty liver disease (NAFLD) is characterized by excess liver fat in the absence of significant alcohol consumption. It can progress from simple steatosis to nonalcoholic steatohepatitis (NASH) and fibrosis and eventually to chronic progressive diseases such as cirrhosis, end-stage liver failure, and hepatocellular carcinoma (*Loomba et al., 2021*). In 2020, an international panel of experts led a consensus-driven process to develop a more appropriate term for the disease utilizing a two-stage Delphi consensus, that is, 'metabolic dysfunction-associated fatty liver disease (MASLD)' related to systemic metabolic dysregulation (*Gofton et al., 2023*; *Rinella et al., 2023*). The pathogenesis of MASLD has not been entirely elucidated. Multifarious factors such as genetic and epigenetic factors, nutritional factors, insulin resistance, lipotoxicity, microbiome, fibrogenesis and hormones secreted from the adipose tissue, are recognized to be involved in the development and progression of MASLD (*Buzzetti et al., 2016*; *Friedman et al., 2018*; *Lee et al., 2017*; *Rada et al., 2020*; *Sakurai et al., 2021*). Free fatty acids (FFAs), which are central to the pathogenesis of MASLD, originate from the periphery, mainly via lipolysis of triglyceride in the adipose tissue, or from increased hepatic de novo lipogenesis (DNL). Fatty acids in hepatocytes undergo mitochondrial β-oxidation and re-esterification to form triglyceride (TG), which are then exported into the blood as very low-density lipoproteins or stored in lipid droplets. Hepatic lipotoxicity occurs when the disposal of fatty acids through β-oxidation or the formation of TG is overwhelmed, which leads to endoplasmic reticulum (ER) stress, oxidative stress and inflammasome activation (*Friedman et al., 2018*). A cluster of differentiation 36/fatty acid translocase (CD36) and cell death-inducing DFF45-like effector proteins A/C (CIDEA/C) are critical for MASLD progression (*Koonen et al., 2007*; *Matsusue et al., 2008*; *Sans et al., 2019*). CD36 can increase FFAs uptake in the liver and drive hepatosteatosis onset. Overexpression of CD36 in hepatocytes increased FFAs uptake and TG storage; conversely, its deletion ameliorated hepatic steatosis and insulin resistance in DIO mice (*Rada et al., 2020*). Additionally, CIDEA/C can also regulate various aspects of lipid homeostasis, including lipid storage, lipolysis, and lipid secretion (*Xu et al., 2024*). As a transcription regulator of *Cd36* and *Cidea/c*, peroxisome proliferator-activated receptor gamma (PPARγ) plays a crucial role in MASLD progression (*Lee et al., 2023b*; *Lee et al., 2021*; *Lee et al., 2018*; *Matsusue et al., 2008*; *Puri et al., 2008*; *Skat-Rørdam et al., 2019*).

Epigenetics, an inheritable phenomenon occurring without altering the DNA sequence, can regulate gene expression through different forms, including DNA methylation, histone modifications, chromatin remodeling, transcriptional control, and non-coding RNAs (*Mann, 2014*). Histone modifications, including acetylation, methylation, phosphorylation, ubiquitination, riboDylation, and ubiquitin-like protein modification (SUMO), are important epigenetic determinants of chromatin tightness and accessibility (*Chen and Pikaard, 1997*). Histone methylation is associated with chromatin-specific transcriptional activity states; for example, methylation of lysine 4 of histone H3 (H3K4), H3K36 and H3K79 are linked with a transcriptional activation state, and H3K9, H3K27, and H4K20 with transcriptional repression state (*Pirola and Sookoian, 2022*). Previous studies have illustrated that epigenetics factors including histone modification play key role in lipid metabolism (*Bayoumi et al., 2020*; *Byun et al., 2020*; *Jun et al., 2012*).

Long non-coding RNAs (lncRNAs) are non-coding RNAs with more than 200 bases in length, can be transcribed by RNA polymerase II, and are comparable to mRNA but lack the crucial open reading framework required for translation (*Ng et al., 2013*; *Ulitsky and Bartel, 2013*). LncRNAs are involved in epigenetic regulation of gene expression at different levels and through different molecular mechanisms such as chromatin remodeling, transcriptional regulation and post-transcriptional processing. Previous studies have indicated that lncRNAs are involved in the pathological progress of MASLD (*Bayoumi et al., 2020*; *Sommerauer and Kutter, 2022*). Although histone modification and lncRNAs influence the susceptibility to MASLD, their roles in MASLD remain largely unknown.

Small nucleolar RNA host genes (SNHG) family, a type of lncRNA, serve as host genes for producing intronic small nucleolar RNAs (snoRNAs) and are mainly related to tumor pathophysiology by regulating proliferation, apoptosis, invasion, and migration (*Sen et al., 2020*; *Zimta et al., 2020*). The family of mouse *Snhg* genes has 19 members including *Snhg1-18*, *Snhg20* and *Snhg7os*. Here, we found that the expression of hepatic *Snhg3* was decreased in high-fat diet (HFD)-induced obesity (DIO) mice. Experiments conducted using in vivo and in vitro models indicated that *Snhg3* was involved in fatty acid metabolism and hepatic steatosis. Mechanistically, *Snhg3* interacted with staphylococcal nuclease and Tudor domain containing 1 (SND1), a well-understood Tudor protein that participates

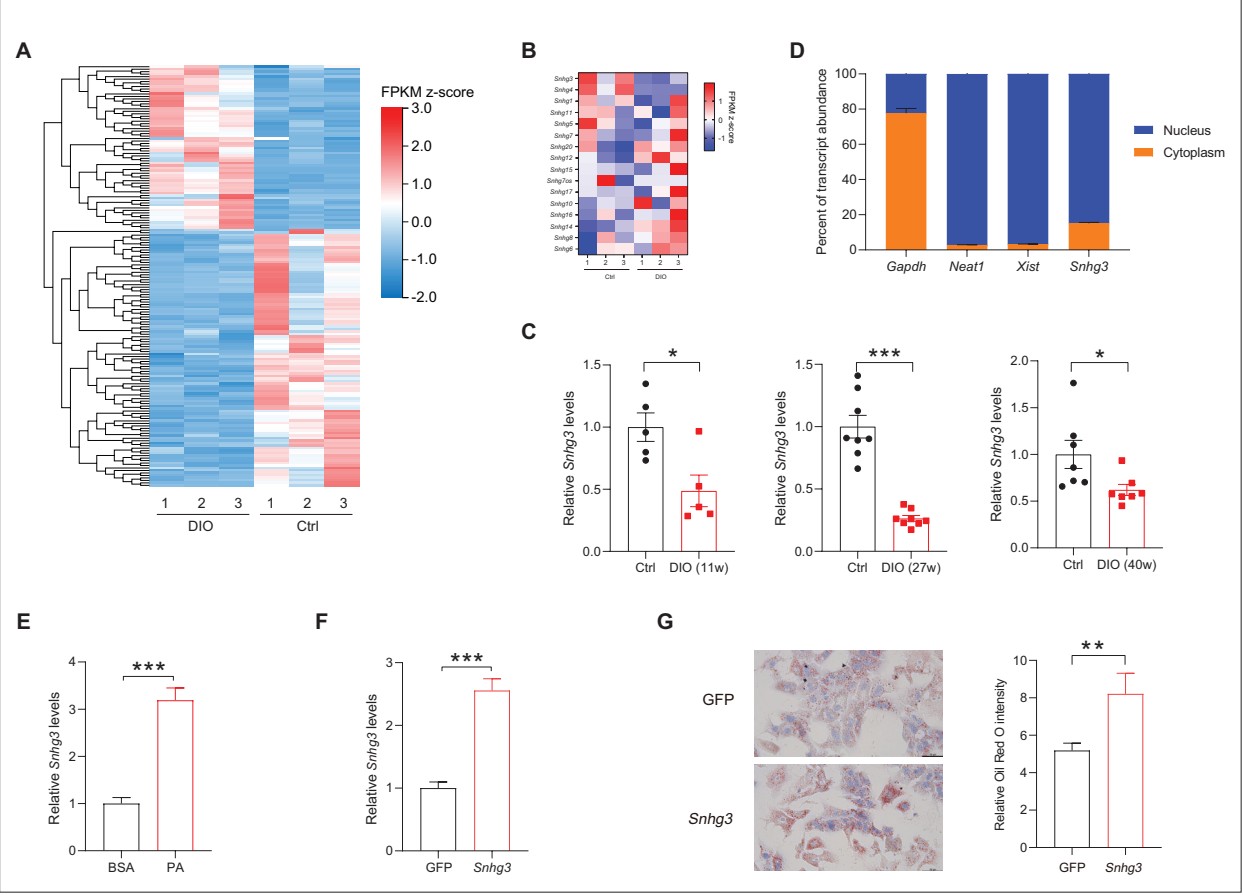

**Figure 1.** The expression of hepatic lncRNA-*Snhg3* is downregulated in DIO mice. (**A**) Differentially expressed *lncRNAs* in livers of 6~8-week-old littermate male mice that were fed an HFD and control diet for 27weeks (n=3mice/group). (**B**) Heat map of *Snhgs* in livers of mice as indicated in (**A**) (n=3mice/group). (**C**) Expression levels of *Snhg3* in the liver of 6~8-week-old littermate male mice that were fed an HFD and control diet for indicated time period 11, 27, and 40weeks. (**D**) Relative *Snhg3* expression levels in nuclear and cytosolic fractions of mouse primary hepatocytes. Nuclear controls: *Neat1* and *Xist*; Cytosolic control: *Gapdh*. (**E**) PA promotes the expression of *Snhg3* in primary hepatocytes. (**F and G**) Overexpression of *Snhg3* (**F**) induces lipid accumulation (**G**) left, Oil red O staining; right, quantitative analysis) in primary hepatocytes with PA treatment. Data are represented as mean ± SEM. *p<0.05, **p<0.01and ***p<0.001 by Student's t test.

The online version of this article includes the following source data for figure 1:

**Source data 1.** The lncRNAs expression profiles in the livers of high-fat diet-induced obesity mice and normal chow-fed mice were determined using RNA-Seq for *Figure 1A*.

in lipid metabolism and tumoral behavior by modulating cholesterol and glycerophospholipid metabolism and acylglyceride storage in lipid droplets (*Navarro-Imaz et al., 2020*). Furthermore, *Snhg3* increased the expression of SND1 by promoting the stability of SND1 mediated by K63-linked ubiquitination and induced nuclear localization of SND1 protein, thereby reducing tri-methylation at H3K27 (H3K27me3) enrichment and boosting chromatin loose remodeling at *Pparg* promoter, eventually enhancing *Pparg*, *Cd36* and *Cidea/c* expressions. Our result indicated that SND1/H3K27me3/PPARγ is partially responsible for *Sngh3*-induced hepatic steatosis.

## Results
### LncRNA-*Snhg3* is downregulated in DIO mice

Firstly, we analyzed the lncRNAs expression profiles in the livers of DIO mice and normal chow-fed mice (control) by RNA-Seq, and found 18072 hepatic lncRNAs, including 338 differentially expressed lncRNAs (q-value ≤0.05, *Figure 1A*). Of all *Snhgs*, *Snhg3* had the most prominent expression and exhibited more noticeable downregulation in the liver of the DIO mice compared to the control mice

(*Figure 1B*), thus, it was selected for further study. The downregulation of *Snhg3* was confirmed by RT-qPCR (*Figure 1C*). Additionally, the Coding Potential Calculator indicated that *Snhg3* has a coding probability of 0.020757, classifying it as a noncoding sequence (*Kang et al., 2017*). Localization of *Snhg3* was primarily observed in nuclei with a probability score of 0.451138, as predicted using software prediction (http://lin-group.cn/server/iLoc-LncRNA/predictor.php). The exact nuclear localization of *Snhg3* was further confirmed by nuclear/cytoplasmic fractionation (*Figure 1D*). Interestingly, the expression of *Snhg3* was induced by palmitic acid (PA) in primary hepatocytes (*Figure 1E*). Furthermore, overexpression of *Snhg3* increased lipid accumulation in primary hepatocytes with PA treatment (*Figure 1F and G*).

## Hepatocyte-specific *Snhg3* knock-out alleviates hepatic steatosis in DIO mice

Given *Snhg3* was associated with hepatic nutrition change, the role of *Snhg3* was further confirmed by constructing hepatocyte-specific *Snhg3* knock-out (*Snhg3*-HKO) mice that were then induced obesity with a high-fat diet (*Figure 2A* and *Figure 2—figure supplement 1A, B*). The result indicated that body weight was mildly decreased in *Snhg3*-HKO mice compared with the control *Snhg3*<sup>*flox/flox*</sup> (*Snhg3*-Flox) mice (*Figure 2B*). The energy consumption is mainly reflected as the sum of energy utilization during internal heat production using comprehensive laboratory animal monitoring system (CLAMS). Heat production showed an increasing trend but was not statistically significant in *Snhg3*-HKO mice (*Figure 2—figure supplement 1C*). Moreover, there were no obvious differences in total oxygen consumption, carbon dioxide production or respiratory exchange ratio (RER) between *Snhg3*-HKO and control mice (*Figure 2—figure supplement 1C*). Furthermore, insulin sensitivity, not glucose tolerance, was improved in *Snhg3*-HKO mice (*Figure 2C*). The *Snhg3*-HKO mice had a decrease in liver weight and the ratio of liver weight/body weight, and improved hepatic steatosis, including decreasing lipid accumulations and the ballooning degeneration of liver cells (*Figure 2D–F*). However, the hepatic fibrosis phenotype showed no difference (*Figure 2—figure supplement 1D*). Serum alanine transaminase (ALT) and aspartate transaminase (AST) levels were significantly decreased in *Snhg3*-HKO mice (*Figure 2G*). Moreover, serum FFAs, TG and TC were also reduced in *Snhg3*-HKO mice (*Figure 2H*). The *Snhg3*-HKO mice exhibited a decrease in inguinal white adipose tissue (iWAT) weight and weight/body weight ratio, while brown adipose (BAT) weight and weight/body weight ratio remained unaltered (*Figure 2—figure supplement 1E*). Additionally, there was no difference in serum insulin between *Snhg3*-HKO mice and control mice (*Figure 2—figure supplement 1F*). These results suggested that hepatic knockout of *Snhg3* improves hepatic steatosis in mice.

## Hepatocyte-specific *Snhg3* knock-in aggravates hepatic steatosis in DIO mice

Furthermore, the hepatocyte-specific *Snhg3* knock-in (*Snhg3*-HKI) mice were also constructed and subsequently induced obesity with a high-fat diet to detect the function of *Snhg3* in the liver (*Figure 3A* and *Figure 3—figure supplement 1A*). The *Snhg3*-HKI mice showed greater weight gains than the control wild type (WT) mice (*Figure 3B*). Insulin sensitivity was also impaired in *Snhg3*-HKI mice (*Figure 3C*). The liver weight and the ratio of liver weight/body weight of *Snhg3*-HKI mice were markedly increased (*Figure 3D*). Also, *Snhg3*-HKI mice exhibited severe hepatic steatosis (*Figure 3E and F*) and higher serum ALT and AST levels (*Figure 3G*). Both serum TC and iWAT weight were increased in *Snhg3*-HKI mice (*Figure 3H* and *Figure 3—figure supplement 1B*). Similar to *Snhg3*-HKO mice, there was also no differences in heat production, total oxygen consumption, carbon dioxide production, RER, hepatic fibrosis phenotype, and serum insulin between *Snhg3*-HKI mice and WT mice (*Figure 3—figure supplement 1C–E*). These findings indicated that upregulation of *Snhg*3 could promote hepatic steatosis.

## *Snhg3* promotes hepatic steatosis by regulating chromatin remodeling

To clarify the molecular mechanism of *Snhg3* in hepatic steatosis, we investigated the hepatic differentially expressed genes (DEGs) using RNA-Seq. There were 1393 DEGs between the *Snhg3*-HKI and control WT mice, with 1028 genes being upregulated and 365 genes downregulated (log2FC ≥1, q-value <0.001) in the liver of *Snhg3*-HKI mice (*Figure 4A*). A gene set enrichment analysis (GSEA) of DEGs revealed that *Snhg3* exerts a global effect on the expression of genes involved in fatty acid

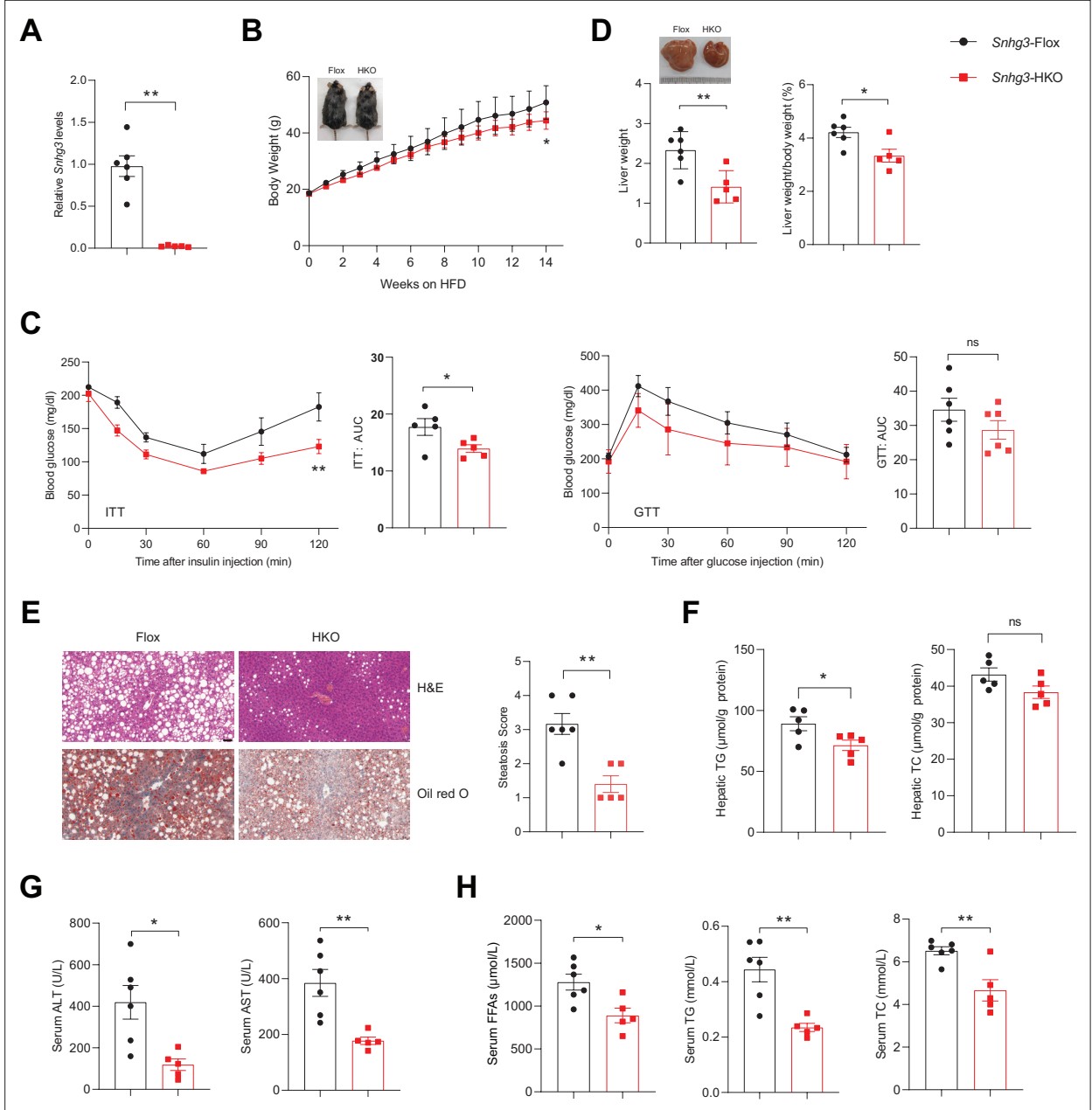

**Figure 2.** Hepatocyte-specific *Snhg3* knockout alleviates hepatic steatosis in DIO mice. (**A**) The expression of *Snhg3* was downregulated in the liver of *Snhg3*-HKO mice. *Snhg3*-Flox (n=6) and *Snhg3*-HKO (n=5). (**B**) Body weights of *Snhg3*-Flox (n=6) and *Snhg3*-HKO (n=5) mice fed HFD for indicated time period. (**C**) ITT (n=5/group) and GTT (n=6/group) of *Snhg3*-Flox and *Snhg3*-HKO mice fed HFD for 18weeks were analyzed, (AUC, Area Under Curve). (**D**) Liver weight (left) and ratio (right) of liver weight/body weight of *Snhg3*-Flox (n=6) and *Snhg3*-HKO (n=5) mice fed HFD for 21weeks. (**E**) H&E and oil red O staining (left) and NASH score (right) of liver of *Snhg3*-Flox and *Snhg3*-HKO mice as indicated in (**D**). Scale bars, 50μm. (**F**) Hepatic TG and TC contents of mice as indicated in (**D**). (**G**) Serum ALT and AST concentrations of mice as indicated in (**D**). (**H**) Serum FFAs, TG and TC concentrations of mice as indicated in (**D**). Data are represented as mean ± SEM. *p<0.05and **p<0.01 by two-way ANOVA (**B and C**) and by Student's t test (the others).

The online version of this article includes the following figure supplement(s) for figure 2:

**Figure supplement 1.** Hepatocyte-specific *Snhg3* knockout alleviates hepatic steatosis in DIO mice.

metabolism and the PPAR signaling pathway (*Figure 4B*). RT-qPCR analysis confirmed that the hepatic expression levels of some genes involved in fatty acid metabolism, including *Cd36*, *Cidea/c*, and stearoyl-CoA desaturase (*Scd1/2*), the key enzymes involved in the biosynthesis of unsaturated fatty acids (*Ntambi and Miyazaki, 2003*), were upregulated in *Snhg3*-HKO mice and were downregulated in *Snhg3*-HKI mice compared to the controls (*Figure 4C*). Additionally, deficiency and overexpression

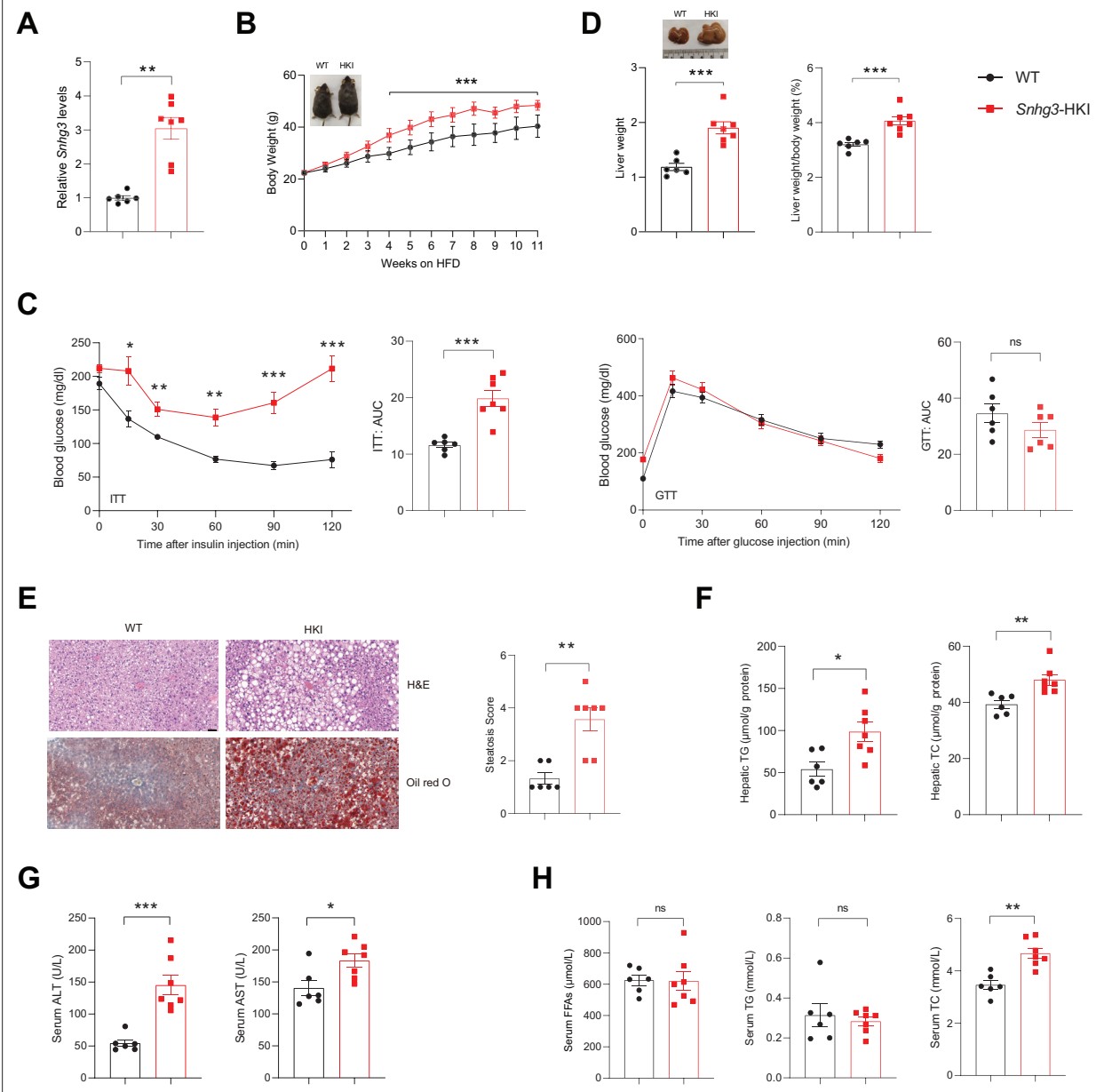

**Figure 3.** Hepatocyte-specific *Snhg3* overexpression aggravates hepatic steatosis in DIO mice. (**A**) The expression of *Snhg3* was upregulated in the liver of *Snhg3*-HKI mice. WT (n=6) and *Snhg3*-HKI (n=7). (**B**) Body weights of WT mice (n=6) and *Snhg3*-HKI mice (n=7) fed HFD for indicated times. (**C**) ITT and GTT of WT (n=6) and *Snhg3*-HKI (n=7) mice fed HFD for 11weeks were analyzed. (**D**) Liver weight (left) and ratio (right) of liver weight/body weight of WT (n=6) and *Snhg3*-HKI (n=7) mice fed HFD for 13weeks. (**E**) Liver H&E and oil red O staining (left) and NASH score (right) of WT and *Snhg3*-HKI mice as indicated in (**D**). Scale bars, 50µm. (**F**) Hepatic TG and TC contents of mice as indicated in (**D**). (**G**) Serum ALT and AST concentrations of mice as indicated in (**D**). (**H**) Serum FFAs, TG and TG concentrations of mice as indicated in (**D**). Data are represented as mean ± SEM. *p<0.05, **p<0.01and ***p<0.001 by two-way ANOVA (**B and C**) and by Student's t test (the others).

The online version of this article includes the following figure supplement(s) for figure 3:

**Figure supplement 1.** Hepatocyte-specific *Snhg3* overexpression aggravates hepatic steatosis in DIO mice.

of *Snhg3* respectively decreased and increased the expression of profibrotic genes, such as collagen type I alpha 1/2 (*Col1a1* and *Col1a2*), but had no effects on the pro-inflammatory factors, including transforming growth factor β1 (*Tgfb1*), tumor necrosis factor a (*Tnfa*), interleukin 6 and 1b (*Il6 and Il1b*; *Figure 4—figure supplement 1A, B*). LncRNAs in the nucleus can affect gene expression in multiple ways, such as chromatin remodeling, transcriptional regulation, and post-transcriptional processing (*Morey and Avner, 2004*; *Thomson and Dinger, 2016*). Since *Snhg3* was mainly localized

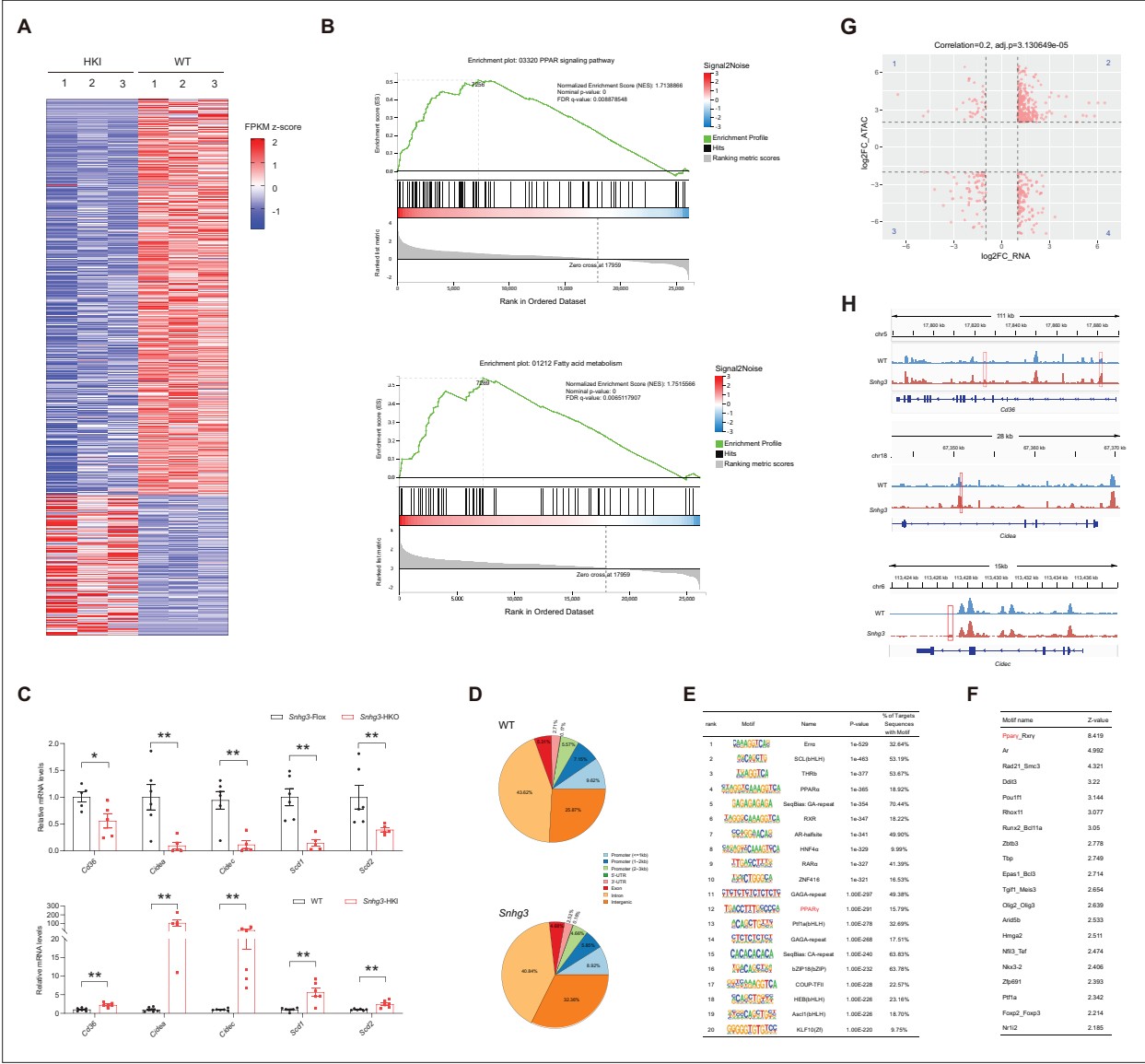

Figure 4. *Snhg3* promotes hepatic steatosis through regulating chromatin remodeling. (**A**) Differentially expressed genes in livers of *Snhg3*-HKI and WT mice (n=3mice/group). (**B**) GSEA showing the enrichment of PPAR signaling pathway (up) and fatty acid metabolism (down) (KEGG pathway database) in livers of *Snhg3*-HKI and WT mice (n=3mice/group). (**C**) Relative hepatic mRNA levels of fatty acid metabolism were measured in *Snhg3*-HKO (up) mice and *Snhg3*-HKI mice (down) compared to the controls. (**D**) Genome distribution ratio of the differentially accessible regions in the liver between WT and *Snhg3*-HKI mice by ATAC-Seq. (**E and F**) The transcription factors analysis in the accessible regions of the liver of *Snhg3*-HKI mice by HOMER (**E**) and CREMA (**F**). (**G**) Integrated ATAC-Seq data with RNA-Seq data. (**H**) Chromatin accessibility at *Cd36* and *Cidea/c* genes. Data are represented as mean ± SD. *p<0.05and **p<0.01 by Student's t test.

The online version of this article includes the following source data and figure supplement(s) for figure 4:

**Source data 1.** The hepatic differentially expressed genes between DIO *Snhg3*-HKI and control WT mice were determined using RNA-Seq for *Figure 4A*.

**Source data 2.** The genome-wide chromatin accessibility in the liver of DIO *Snhg3*-HKI and WT mice was determined using ATAC-Seq, related to *Figure 4D*.

**Source data 3.** The genes were associated specifically with the differentially accessible regions in genome in the liver between DIO *Snhg3*-HKI and WT mice, related to *Figure 4D*.

**Source data 4.** The hepatic differentially expressed genes between DIO *Snhg3*-HKI and WT mice were correlated with open chromatin regions by integrated analyzing ATAC-Seq data with RNA-Seq data for *Figure 4G*.

**Figure supplement 1.** Snhg3 influences the expression of profibrotic genes, not pro-inflammatory factors.

in the nuclei of hepatocytes, we next checked the genome-wide chromatin accessibility (log2FC >2, p-value <0.001) in the liver of *Snhg3*-HKI and WT mice using ATAC-Seq. We discovered that in all 6810 differentially accessible regions (DARs), 4305 (>63.2%) were more accessible in *Snhg3*-HKI mice and only 2505 (>36.8%) of peaks were more accessible in control mice, indicating that the chromatin states were 'hyper-accessible' in the liver of *Snhg3*-HKI mice. Moreover, DARs were with relatively few promoter-proximal (Up2k) and exon regions in both the control and *Snhg3*-HKI groups (*Figure 4D*), supporting the idea that gene activation depends on multiple regulatory regions, is not limited to its promoter and exon regions (*Ackermann et al., 2016*). Furthermore, 3966 genes were associated specifically with the accessible regions in the *Snhg3*-HKI group and only 2451 genes in the WT group (log2FC >2, p-value <0.001). Additionally, *PPARg* was identified as a potential transcription factor associated with hyper-accessible regions in the liver of the *Snhg3*-HKI group by HOMER and CREMA (*Figure 4E and F*).

To determine whether open chromatin regions were correlated with gene expression, we integrated ATAC-Seq data (genes associated with DARs, log2FC >2, p-value <0.001) with RNA-Seq data (DEGs in DIO *Snhg3*-HKI and control mice, log2FC >1, q-value <0.001). Overall, 233 upregulated genes shown in quadrant 2, including *Cd36* and *Cidea/c*, had at least one associated open chromatin region, which accounted for >22.67% (total 1028) of DEGs mapped to ATAC-Seq peaks in the liver of *Snhg3*-HKI mice (*Figure 4G and H*). Meanwhile, at least one open chromatin region was associated with 65 downregulated genes in the quadrant 3, which accounted for >17.81% (total 365) of DEGs mapped to ATAC-Seq peaks in the liver of WT mice (*Figure 4G*).

## *Snhg3* induces SND1 expression by interacting with SND1 and enhancing the stability of SND1 protein

To further elucidate the molecular mechanism of *Snhg3* in hepatic steatosis, an RNA pull-down followed by mass spectrometry (MS) assay was performed in primary hepatocytes. The result identified 234 specific *Snhg3*-associated proteins, involved in multiple signaling pathways, including PPAR, NAFLD and fatty acid degradation pathways (*Figure 5A and B*). Of these proteins, a well-understood Tudor protein SND1 was also predicted to interact with three fragments of *Snhg3* by bioinformatic method (RBPsuite; *Figure 5C and D*). *Snhg3* coprecipitation with SND1 was confirmed by RNA pull-down coupled with western blotting (*Figure 5E*), which was consistent with the RNA immunoprecipitation (RIP) assay results (*Figure 5F*). Meanwhile, *Snhg3* regulated the protein, not mRNA, expression of SND1 in vivo and in vitro by mildly promoting the stability of SND1 protein (*Figure 5G–J*). Furthermore, we tested the effect of *Snhg3* on the ubiquitin-modification of SND1 and found that *Snhg3* enhanced SND1 ubiquitination in vivo and in vitro (*Figure 5K and L*). Previous studies indicated that K48-linked polyubiquitination aids in proteasome-mediated recognition and degradation and that K63-linked polyubiquitination participates in signaling assemblies and protein stability (*Sun et al., 2020*). As predicted, *Snhg3* overexpression increased K63-linked ubiquitination modification in endogenous and exogenous SND1 protein, not K48- or K33-linked (*Figure 5M and N*). Additionally, *Snhg3* overexpression enhanced the nuclear localization of SND1 in Hepa1-6 cells with PA treatment (*Figure 5O*). Collectively, these results suggested that *Snhg3* promoted the K63-linked ubiquitination and stability of SND1 protein through interacting with SND1, thus resulting in SND1 protein increase and nuclear localization.

## *Snhg3* promotes PPARγ expression by decreasing H3K27me3 enrichment at the *Pparg* promoter

SND1, initially named as p100, is a highly conserved and ubiquitously expressed multifunctional Tudor domain-containing protein that participates in pivotal biological processes like double-stranded RNA editing, pre-mRNA splicing, microRNA-mediating gene silencing and piRNA biogenesis in germlines (*Ying and Chen, 2012*). Previous studies indicated that Tudor proteins participate in epigenetic regulation by binding to methyl-arginine/lysine residues (*Ying and Chen, 2012*). However, whether SND1 influences histone modification remains unclear. It is well known that histone modification dynamically regulates specific gene expression by altering the organization and function of chromatin and is involved in the pathophysiology of some diseases, such as histone H3 methylation modification, which may contribute to MASLD pathogenesis (*Byun et al., 2020*; *Jun et al., 2012*; *Tessarz and Kouzarides, 2014*). Considering that H3K27me3, a repressive chromatin mark, plays a role in autophagy-mediated

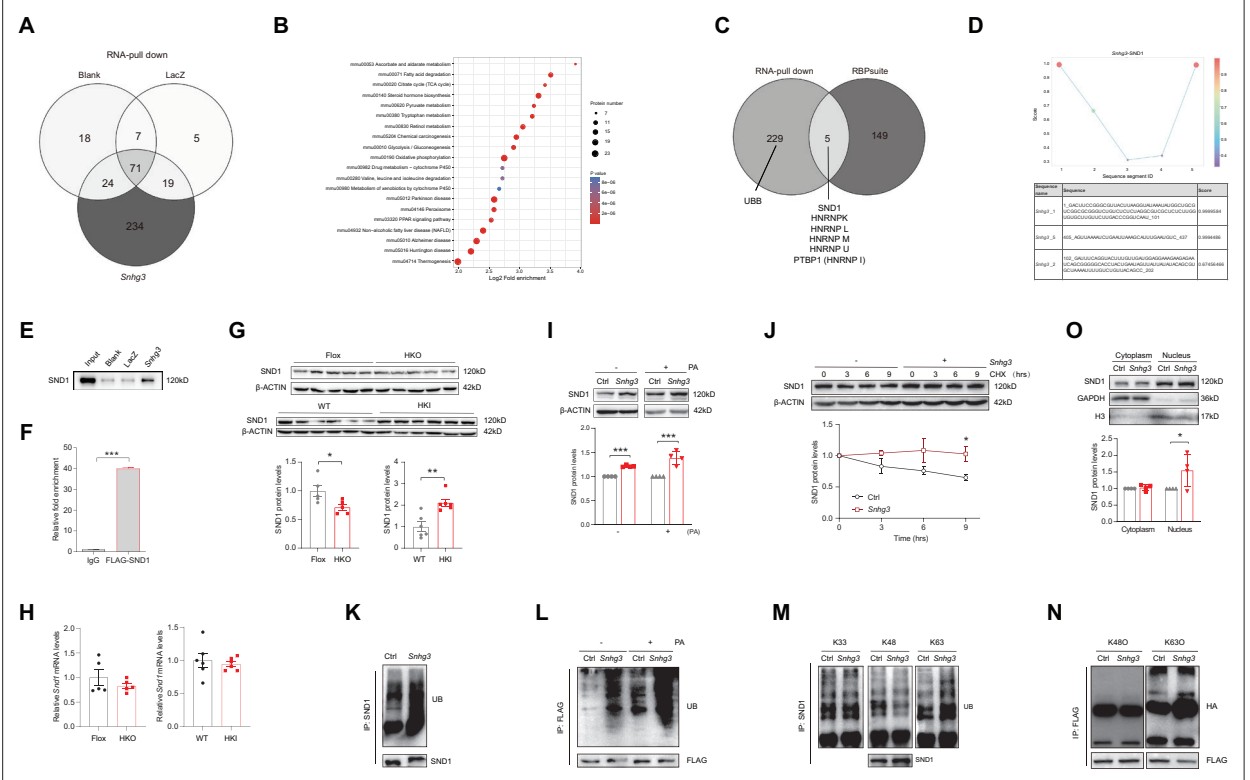

**Figure 5.** *Snhg3* induces SND1 expression and enhances the stability of SND1 protein through physiologically interacting with SND1. (**A**) Venn diagram of data from RNA pull-down and MS. (**B**) KEGG analysis of genes in specific *Snhg3*-binding proteins from RNA pull-down and MS. (**C**) Venn diagram of data from RNA pull-down and MS and bioinformatics predicted by RBPsuite. (**D**) SND1 interacts with different fragments of *Snhg3* predicted by bioinformatics using RBPsuite. (**E**) RNA pull-down and western blotting confirms *Snhg3* interacting with SND1. (**F**) RIP confirms SND1 interacting with *Snhg3*. (**G and H**) Relative protein (**G**, up, western blotting; down, quantitative result) and RNA (**H**) levels of *Snd1* were measured in the liver. (**I**) *Snhg3* enhanced the protein level of SND1 in Hepa1-6 cells (up, western blotting; down, quantitative result). (**J**) *Snhg3* promoted the stability of SND1 protein in Hepa1-6 cells (up, western blotting; down, quantitative result). (**K and L**) *Snhg3* promoted the ubiquitination of endogenous (**K**) and exogenous (**L**) SND1 protein in Hepa1-6 cells. (**M and N**) *Snhg3* increased the K63-linked, not K48-linked and K33-linked, ubiquitination modification of endogenous (**M**) and exogenous (**N**) SND1 protein. (**O**) *Snhg3* induced the nuclear localization of SND1 in Hepa1-6 cells (up, western blotting; down, quantitative result). Data are represented as mean ± SEM. *p<0.05 and ***p<0.001 by two-way ANOVA (**J**) or Student's t test (the others).

The online version of this article includes the following source data for figure 5:

**Source data 1.** *Snhg3*-bound proteins were identified in mouse primary hepatocytes by RNA-Pulldown-Mass spectrometry for *Figure 5A*.

**Source data 2.** *Snhg3*-bound proteins were predicted by bioinformatic method (RBPsuite) for *Figure 5C*.

**Source data 3.** PDF file containing original western blots for *Figure 5E*, indicating the relevant bands and treatments.

**Source data 4.** Original files for western blot analysis displayed in *Figure 5E*.

**Source data 5.** PDF file containing original western blots for *Figure 5G*, indicating the relevant bands and treatments.

**Source data 6.** Original files for western blot analysis displayed in *Figure 5G*.

**Source data 7.** PDF file containing original western blots for *Figure 5I*, indicating the relevant bands and treatments.

**Source data 8.** Original files for western blot analysis displayed in *Figure 5I*.

**Source data 9.** PDF file containing original western blots for *Figure 5J*, indicating the relevant bands and treatments.

**Source data 10.** Original files for western blot analysis displayed in *Figure 5J*.

**Source data 11.** PDF file containing original western blots for *Figure 5K*, indicating the relevant bands and treatments.

**Source data 12.** Original files for western blot analysis displayed in *Figure 5K*.

**Source data 13.** PDF file containing original western blots for *Figure 5L*, indicating the relevant bands and treatments.

**Source data 14.** Original files for western blot analysis displayed in *Figure 5L*.

**Source data 15.** PDF file containing original western blots for *Figure 5M*, indicating the relevant bands and treatments.

**Source data 16.** Original files for western blot analysis displayed in *Figure 5M*.

*Figure 5 continued on next page*

*Figure 5 continued*
**Source data 17.** PDF file containing original western blots for *Figure 5N*, indicating the relevant bands and treatments.

**Source data 18.** Original files for western blot analysis displayed in *Figure 5N*.

**Source data 19.** PDF file containing original western blots for *Figure 5O*, indicating the relevant bands and treatments.

**Source data 20.** Original files for western blot analysis displayed in *Figure 5O*.

lipid degradation (*Byun et al., 2020*), we tested the effect of SND1 on H3K27me3. The results revealed that both SND1 and *Snhg3* overexpression reduced the H3K27me3 level (*Figure 6A*). Furthermore, disrupting SND1 expression increased the H3K27me3 level and reversed the *Snhg3*-induced H3K27me3 decrease (*Figure 6B and C*). Moreover, the hepatic H3K27me3 level was upregulated in *Snhg3*-HKO mice but downregulated in *Snhg3*-HKI mice (*Figure 6D*). The results indicated that *Snhg3* negatively regulated the H3K27me3 level through SND1.

To further clarify whether *Snhg3*-induced H3K27me3 decrease is involved in hepatic steatosis, we examined the H3K27me3 enrichment in the liver of *Snhg3*-HKO mice using the CUT&Tag assay and detected 10915 peaks. The genomic locations of these peaks were divided into eight categories, and the H3K27me3 signals were predominantly enriched (about 54%) at the 2 kb promoter, 5'-untranslated region (5'-UTR), and exon categories. Meanwhile, very few signals (about 14%) were enriched in the 2 kb downstream and intergenic categories in the liver of *Snhg3*-HKO mice (*Figure 6E*). Moreover, the exon, upstream 2 k, 5'-UTR and intron regions of *Pparg* were enriched with the H3K27me3 mark (fold_enrichment = 4.15697) in the liver of *Snhg3*-HKO mice. Subsequently, ChIP assay revealed that hepatic H3K27me3 enrichment at the *Pparg* promoter was increased in *Snhg3*-HKO mice but decreased in *Snhg3*-HKI mice (*Figure 6F*). *Snhg3*-overexpression in Hepa1-6 cells yielded similar results (*Figure 6G*).

## SND1 mediates *Snhg3*-induced PPARγ upregulation

PPARγ has been reported to influence MASLD progression by regulating target genes such as *Cd36* and *Cidea/c* (*Lee et al., 2023b*; *Lee et al., 2021*; *Lee et al., 2018*; *Matsusue et al., 2008*; *Skat-Rørdam et al., 2019*). In this study, the mRNA and protein expression levels of hepatic PPARγ were decreased in *Snhg3*-HKO mice and increased in *Snhg3*-HKI mice (*Figure 7A–C*). Additionally, CD36 protein level was decreased in *Snhg3*-HKO mice and increased in *Snhg3*-HKI mice (*Figure 7B and C*). The upregulation of *Snhg3* and SND1 also increased the expression of *Pparg* and *Cd36* in vitro (*Figure 7D–F*). Meanwhile, disruption of SND1 expression alleviated *Snhg3*-induced PPARγ increase and lipid accumulation (*Figure 7G–I*). Collectively, these results demonstrated that SND1 mediated *Snhg3*-induced PPARγ and CD36 expression.

In addition, *Snhg3* serves as host gene for producing intronic U17 snoRNAs, the H/ACA snoRNA. A previous study found that cholesterol trafficking phenotype was not due to reduced *Snhg3* expression, but rather to haploinsufficiency of U17 snoRNA. Upregulation of hypoxia-upregulated mitochondrial movement regulator (HUMMR) in U17 snoRNA-deficient cells promoted the formation of ER-mitochondrial contacts, resulting in decreasing cholesterol esterification and facilitating cholesterol trafficking to mitochondria (*Jinn et al., 2015*). Additionally, disruption of U17 snoRNA caused resistance to lipid-induced cell death and general oxidative stress in cultured cells. Furthermore, knockdown of U17 snoRNA in vivo protected against hepatic steatosis and lipid-induced oxidative stress and inflammation (*Sletten et al., 2021*). In this study, the expression of U17 snoRNA decreased in the liver of *Snhg3*-HKO mice and unchanged in the liver of *Snhg3*-HKI mice, but overexpression of U17 snoRNA had no effect on the expression of SND1 and PPARγ (*Figure 7—figure supplement 1A–C*), indicating that *Sngh3* induced hepatic steatosis was independent on U17 snoRNA.

## PPARγ mediates *Snhg3*-induced hepatic steatosis

Hepatocyte-specific depletion of PPARγ is known to protect mice against NASH and boost the therapeutic efficacy of rosiglitazone, a synthetic PPARγ agonist, in the liver (*Lee et al., 2021*). Furthermore, PPARγ is an inducer of adipocyte differentiation and a reservoir for excess FFAs, thereby potentially preventing lipotoxicity in other tissues and organs (*Medina-Gomez et al., 2007*). To this end, we tested the effect of T0070907, a selective PPARγ antagonist, on *Snhg3*-induced hepatic steatosis in mice. The result showed that T0070907 treatment for 8 weeks had no effects on body weight, liver and

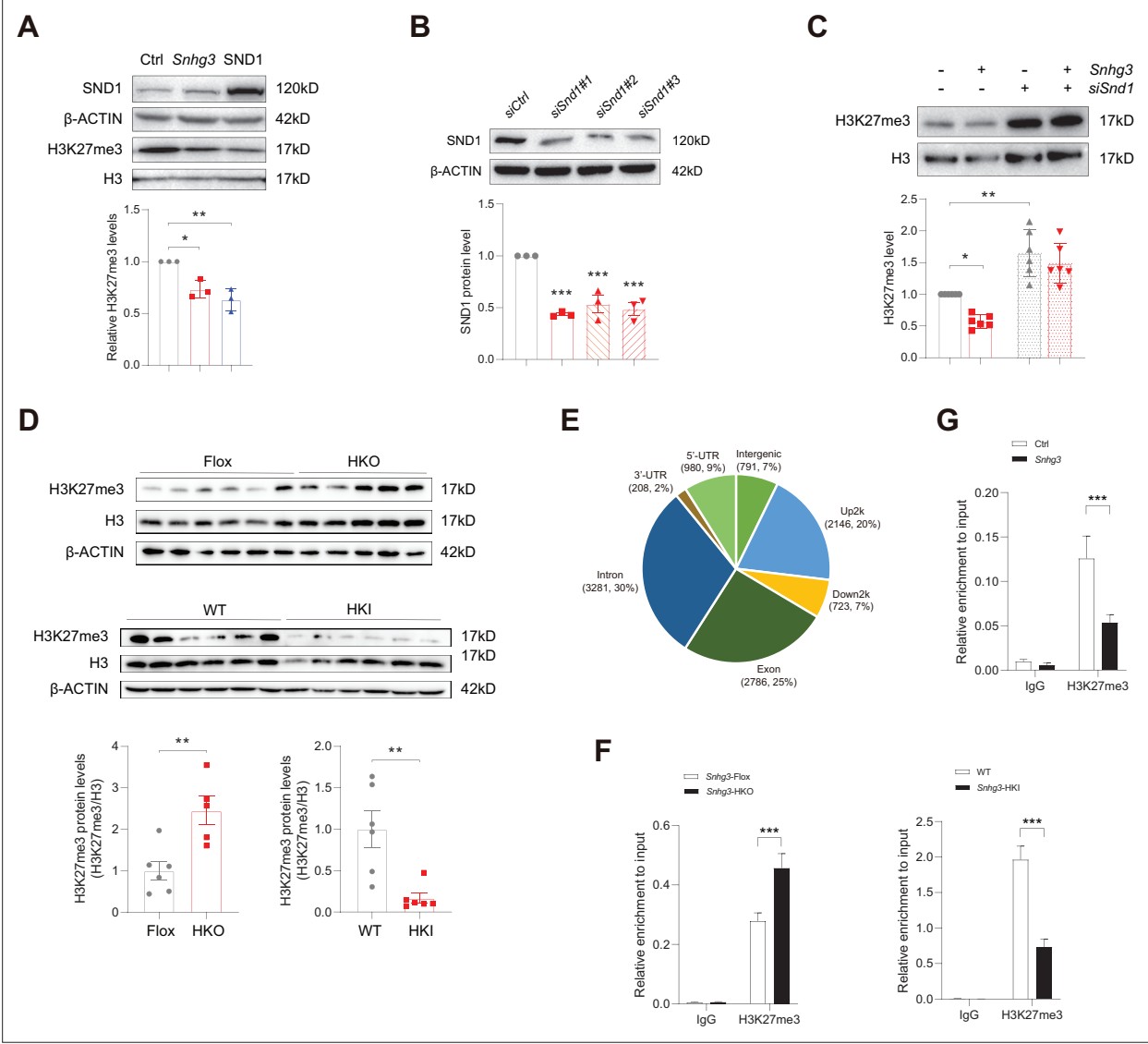

**Figure 6.** *Snhg3* increases PPARγ expression through reducing H3K27me3 enrichment at *Pparg* promoter. (**A**) Overexpression of *Snhg3* or SND1 reduced the H3K27me3 level in Hepa1-6 cells with PA treatment (up, western blotting; down, quantitative result). (**B**) The expression of SND1 was disrupted with siRNA (up, western blotting; down, quantitative result). (**C**) Disruption SND1 expression reversed the *Snhg3*-induced decrease in H3K27me3 in primary hepatocytes (up, western blotting; down, quantitative result). (**D**) The H3K27me3 levels were measured in the liver of *Snhg3*-HKO and *Snhg3*-HKI mice (up, western blotting; down, quantitative result). (**E**) Genome distribution ratio of H3K27me3 enrichment genetic sequence in the liver of *Snhg3*-HKO mice. (**F and G**) ChIP result showed that *Snhg3* affected H3K27me3 enrichment at *Pparg* promoter in vivo (**F**) and in vitro. (**G**) Data are represented as mean ± SEM. *p<0.05, **p<0.01and ***p<0.001 by one-way ANOVA (**C**) or by Student's t test (the others).

The online version of this article includes the following source data for figure 6:

**Source data 1.** PDF file containing original western blots for *Figure 6A*, indicating the relevant bands and treatments.

**Source data 2.** Original files for western blot analysis displayed in *Figure 6A*.

**Source data 3.** PDF file containing original western blots for *Figure 6B*, indicating the relevant bands and treatments.

**Source data 4.** Original files for western blot analysis displayed in *Figure 6B*.

**Source data 5.** PDF file containing original western blots for *Figure 6C*, indicating the relevant bands and treatments.

**Source data 6.** Original files for western blot analysis displayed in *Figure 6C*.

**Source data 7.** PDF file containing original western blots for *Figure 6D*, indicating the relevant bands and treatments.

**Source data 8.** Original files for western blot analysis displayed in *Figure 6D*.

**Source data 9.** The H3K27me3 enrichment in the genome in the liver of DIO *Snhg3*-HKO mice were determined using the CUT&Tag-Seq, related to *Figure 6E*.

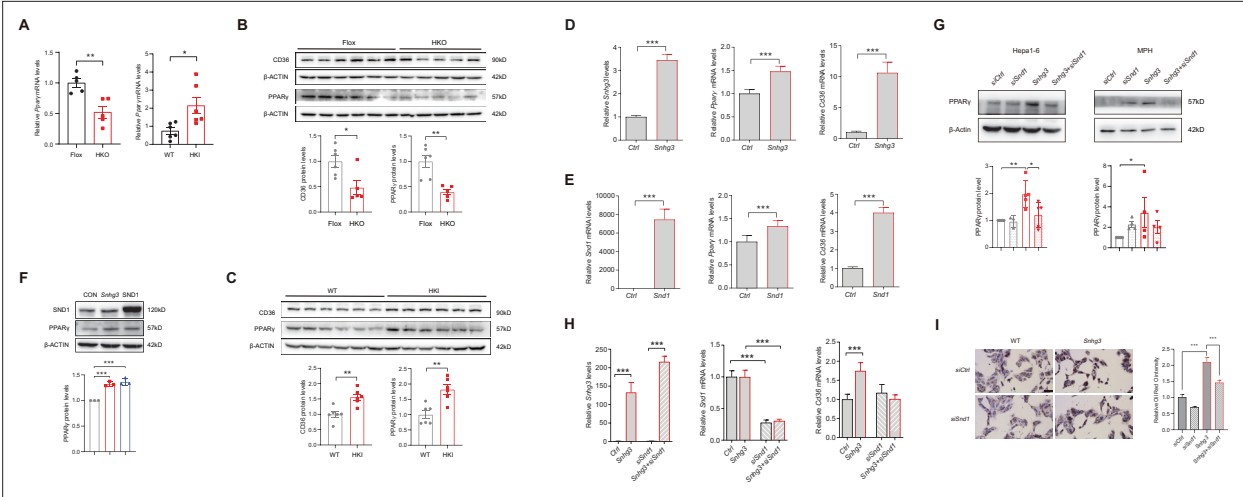

**Figure 7.** SND1 mediates *Snhg3*-induced PPARγ upregulation. (**A**) The mRNA level of *Pparg* was measured in the liver of *Snhg3*-HKO (left) and *Snhg3*-HKI mice (right). (**B**) The protein level of PPARγ was measured in the liver of *Snhg3*-Flox and *Snhg3*-HKO mice (up, western blotting; down, quantitative result). (**C**) The protein level of PPARγ were measured in the liver of WT and *Snhg3*-HKI mice (up, western blotting; down, quantitative result). (**D and E**) Overexpression of *Snhg3* (**D**) and SND1 (**E**) promoted the mRNA expression of *Pparg* and *Cd36* in primary hepatocytes. (**F**) Overexpression of *Snhg3* and SND1 increased the protein expression of PPARγ in Hepa1-6 cells (up, western blotting; down, quantitative result). (**G**) Disruption SND1 expression alleviated *Snhg3*-induced increase in the protein level of PPARγ in Hepa1-6 cells (left) and mouse primary hepatocytes (MPH, right) with PA treatment (up, western blotting; down, quantitative result). (**H**) Disruption SND1 expression alleviated *Snhg3*-induced increase in the mRNA levels of *Pparg* and *Cd36* in Hepa1-6 cells with PA treatment. (**I**) Disruption SND1 expression alleviated *Snhg3*-induced increase in lipid accumulation (left, oil red O staining; right, quantitative result) in MPH with PA treatment. Data are represented as mean ± SEM. *p<0.05, **p<0.01and ***p<0.001 by one-way ANOVA (**G–I**) or by Student's t test (the others).

The online version of this article includes the following source data and figure supplement(s) for figure 7:

**Source data 1.** PDF file containing original western blots for *Figure 7B*, indicating the relevant bands and treatments.

**Source data 2.** Original files for western blot analysis displayed in *Figure 7B*.

**Source data 3.** PDF file containing original western blots for *Figure 7C*, indicating the relevant bands and treatments.

**Source data 4.** Original files for western blot analysis displayed in *Figure 7C*.

**Source data 5.** PDF file containing original western blots for *Figure 7F*, indicating the relevant bands and treatments.

**Source data 6.** Original files for western blot analysis displayed in *Figure 7F*.

**Source data 7.** PDF file containing original western blots for *Figure 7G*, indicating the relevant bands and treatments.

**Source data 8.** Original files for western blot analysis displayed in *Figure 7G*.

**Figure supplement 1.** *Sngh3*-induced changes in PPARγ and SND1 are independent on U17 snoRNA.

**Figure supplement 1—source data 1.** PDF file containing original western blots for *Figure 7—figure supplement 1C*, indicating the relevant bands and treatments.

**Figure supplement 1—source data 2.** Original files for western blot analysis displayed in *Figure 7—figure supplement 1C*.

iWAT weight, and serum FFAs, TG and TC in *Snhg3*-HKI mice, but improved *Snhg3*-induced hepatic steatosis in *Snhg3*-HKI mice (**Figure 8A–D** and **Figure 8—figure supplement 1**). Moreover, T0070907 mitigated the hepatic *Cd36* and *Cidea/c* increase in *Snhg3*-HKI mice (**Figure 8E**). Additionally, *Snhg3*- and SND1-induced *Cd36* increase also were abolished by T0070907 in hepa1-6 cells (**Figure 8F**). Collectively, these results suggested that PPARγ-mediated *Snhg3*-induced hepatic steatosis.

## Discussion

Liver steatosis is common in various metabolic diseases and related disorders, including MASLD. Although lncRNAs are implicated in regulating numerous mechanisms related to liver steatosis and MASLD, their exact function remains to be determined. In this study, lncRNA-*Snhg3* is downregulated in DIO mice and hepatocyte-specific *Snhg3* deficiency improved hepatic steatosis and insulin resistance, while overexpression aggravated hepatic steatosis and insulin resistance in mice. Our results showed that the expression of *Snhg3* was decreased in DIO mice which led us to speculate that

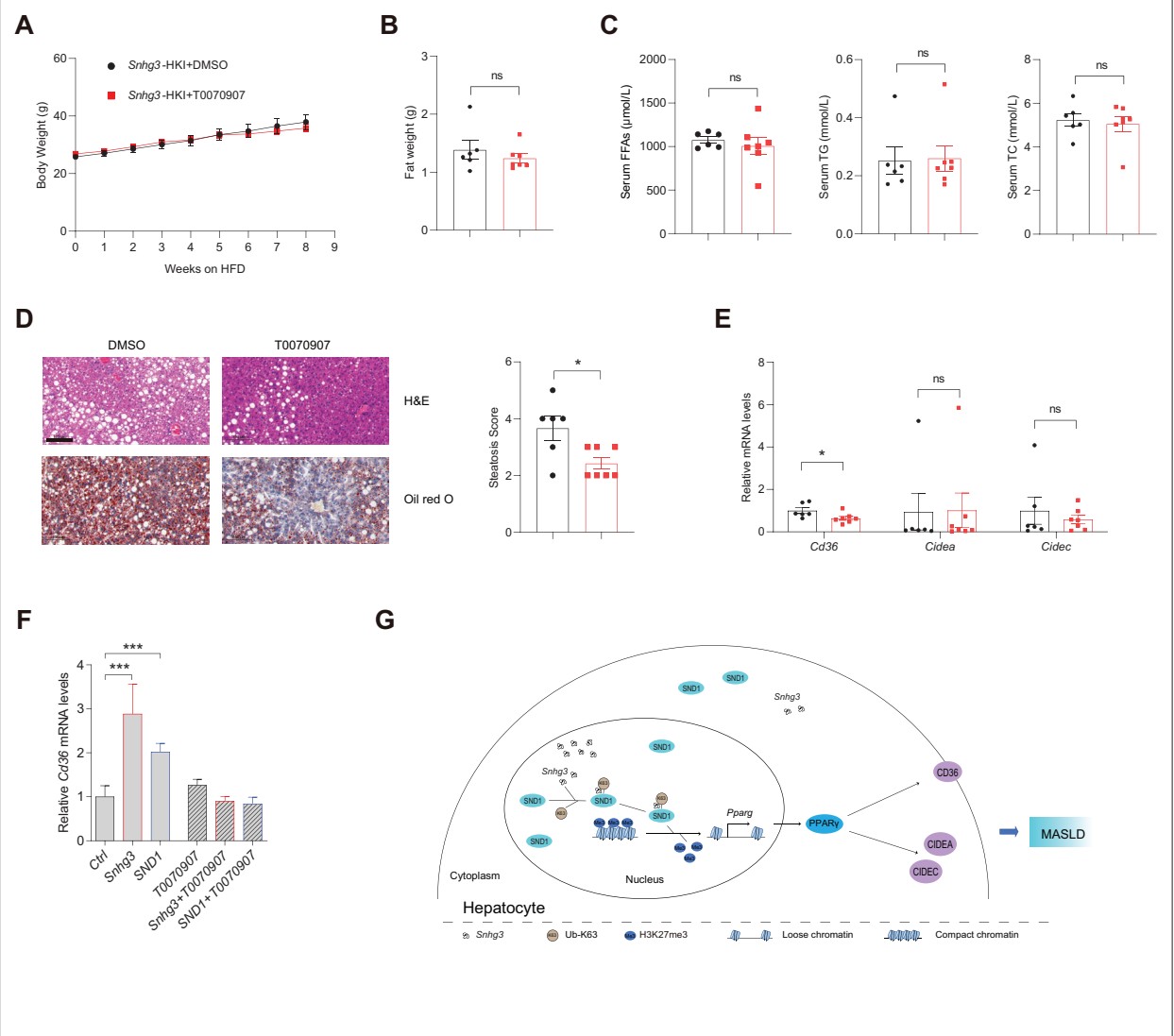

**Figure 8.** PPARγ mediates *Snhg3*-induced hepatic steatosis. (**A and B**) Body weight (**A**) and liver weight (**B**) of *Snhg3*-HKI mice without (n=6) or with (n=7) T0070907 treatment for 8weeks. (**C**) Serum FFAs, TG and TG concentrations of mice as indicated in (**A**). (**D**) Hepatic H&E and oil red O staining (left) and NASH score (right) of mice as indicated in *A*. Scale bars, 100μm. (**E**) T0070907 mitigated the hepatic *Cd36* and *Cidea/c* increase in *Snhg3*-HKI mice. (**F**) T0070907 disrupted *Snhg3*- and SND1-induced *Cd36* increase in Hepa1-6 cells. (**G**) Model of how *Snhg3* and SND1 interacting and influencing chromatin remodeling via H3K27me3, and promoting PPARγ expression thereby resulting in hepatic steatosis. Data are represented as mean ± SEM. *p<0.05and ***p<0.001 by two-way ANOVA (**A**) or by Student's t test for the others.

The online version of this article includes the following figure supplement(s) for figure 8:

**Figure supplement 1.** Fat weight of *Snhg3*-HKI mice without (n=6) or with (n=7) T0070907 treatment for 8weeks.

the downregulation of *Snhg3* might be a stress protective reaction to high nutritional state, but the specific details need to be clarified. This is probably similar to fibroblast growth factor 21 (FGF21) and growth differentiation factor 15 (GDF15), whose endogenous expression and circulating levels are elevated in obese humans and mice despite their beneficial effects on obesity and related metabolic complications (**Keipert and Ost, 2021**). Although FGF21 can be induced by oxidative stress and be activated in obese mice and in NASH patients, elevated FGF21 paradoxically protects against oxidative stress and reduces hepatic steatosis (**Tillman and Rolph, 2020**).

Excessive hepatic lipid deposition owing to increased FFAs uptake and hepatic DNL impairs autophagy and promotes ER stress and oxidative stress, insulin resistance, inflammation, and liver tissue damage, ultimately aggravating MASLD progression (**Rada et al., 2020**). In this study, *Snhg3* induced the expression of fatty acid metabolism related genes such as *Cd36*, *Cidea/c* and *Scd1/2*. Under

physiological conditions, CD36 expression in hepatocytes was found to be minimal; however, lipid overload or activation of nuclear receptors including PPARα/γ and liver X receptor (LXR), could significantly increase it (*Rada et al., 2020*). As a transcription regulator of *Cd36* and *Cidea/c*, it is well known that PPARγ plays major adipogenic and lipogenic roles in adipose tissue. Although the expression of PPARγ in the liver is very low under healthy conditions, induced expression of PPARγ in both hepatocytes and non-parenchymal cells (Kupffer cells, immune cells, and hepatic stellate cells [HSCs]) in the liver has a crucial role in the pathophysiology of MASLD (*Chen et al., 2023*; *Gross et al., 2017*; *Lee et al., 2023b*). The activation of PPARγ in the liver induces the adipogenic program to store fatty acids in lipid droplets as observed in adipocytes (*Lee et al., 2018*). Moreover, the inactivation of liver PPARγ abolished rosiglitazone-induced an increase in hepatic TG and improved hepatic steatosis in lipoatrophic AZIP mice (*Gavrilova et al., 2003*). Furthermore, there is a strong correlation between the onset of hepatic steatosis and hepatocyte-specific PPARγ expression. Clinical trials have also indicated that increased insulin resistance and hepatic PPARγ expressions were associated with NASH scores in some obese patients (*Lee et al., 2023a*; *Mukherjee et al., 2022*). Even though PPARγ's primary function is in adipose tissue, patients with MASLD have much higher hepatic expression levels of PPARγ, reflecting the fact that PPARγ plays different roles in different tissues and cell types (*Mukherjee et al., 2022*). As these studies mentioned above, our result also hinted at the importance of PPARγ in the pathophysiology of MASLD. *Snhg3* deficiency or overexpression respectively induced the decrease or increase in hepatic PPARγ. Moreover, administration of PPARγ antagonist T0070907 mitigated the hepatic *Cd36* and *Cidea/c* increase and improved *Snhg3*-induced hepatic steatosis. However, conflicting findings suggest that the expression of hepatic PPARγ is not increased as steatosis develops in humans and in clinical studies and that PPARγ agonists administration did not aggravate liver steatosis (*Gross et al., 2017*). Thus, understanding how the hepatic PPARγ expression is regulated may provide a new avenue to prevent and treat the MASLD (*Lee et al., 2018*).

Hepatotoxicity accelerates the development of progressive inflammation, oxidative stress, and fibrosis (*Roehlen et al., 2020*). Chronic liver injury including MASLD can progress to liver fibrosis with the formation of a fibrous scar. Injured hepatocytes can secrete fibrogenic factors or exosomes containing miRNAs that activate HSCs, the major source of the fibrous scar in liver fibrosis (*Kisseleva and Brenner, 2021*). Apart from promoting lipogenesis, PPARγ has also a crucial function in improving inflammation and fibrosis (*Chen et al., 2023*). In this study, no hepatic fibrosis phenotype was seen in *Snhg3*-HKO and *Snhg3*-HKI mice. Moreover, the expression levels of profibrotic genes including *Col1a1* and *Col1a2* were decreased in *Snhg3*-HKO mice and increased in *Snhg3*-HKI mice, but the proinflammatory factors including *Tgfb1, Tnfa, Il6, and Il1b* had no changes. Inflammation is an absolute requirement for fibrosis because factors from injured hepatocytes alone are not sufficient to directly activate HSCs and lead to fibrosis (*Kisseleva and Brenner, 2021*). Additionally, previous studies indicated that exposure to HFD for more 24 weeks causes less severe fibrosis (*Alshawsh et al., 2022*). In future, the effect of *Snhg3* on hepatic fibrosis in mice need to be elucidated by prolonged high-fat diet feeding or adopting methionine- and choline deficient diet (MCD) feeding.

Epigenetics plays a crucial role in many physiological and pathological situations (*Peixoto et al., 2020*). Epigenetic regulation induces phenotypic changes that may respond to environmental cues through DNA methylation and histone modification, chromatin remodeling, and noncoding RNAs (*Mann, 2014*). Epigenetic changes interact with inherited risk factors to modulate the individual risk of MASLD development and the severity of progression. Epigenetic modifications, including DNA methylation, miRNAs, and histone modifications, have been associated with MASLD (*Baffy, 2015*; *Eslam et al., 2018*; *Jonas and Schürmann, 2021*). To date, there is no approved pharmacologic therapy for MASLD, and the mainstay of management remains lifestyle changes with exercise and dietary modifications (*Bayoumi et al., 2020*). Therefore, understanding the epigenetic modifications in MASLD pathogenesis might prove a rational strategy to prevent the disease and develop novel therapeutic interventions (*Sodum et al., 2021*).

LncRNAs, being abundant in the genome participate in regulating the expression of coding genes through various molecular mechanisms, including: (1) transcriptional regulation at the promoter of target genes; (2) inhibiting RNA polymerase II or mediating chromatin remodeling and histone modification; (3) interfering with the splicing and processing of mRNA or producing endogenous siRNA; (4) regulating the activity or cellular localization of the target protein; (5) acting as competitive endogenous RNAs; and (6) riboregulation by forming nucleic acid-protein complex as structural component

(*Morey and Avner, 2004*; *Sommerauer and Kutter, 2022*; *Thomson and Dinger, 2016*). However, compared to the large number of lncRNAs, only few have been functionally well-characterized. Collective literature has shown that lncRNAs play a crucial role in MASLD (*Sommerauer and Kutter, 2022*). This study demonstrated that lncRNA-*Snhg3* participated in the pathology of MASLD by epigenetic modification; that is, *Snhg3* inhibited the H3K27me3 level, and promoted chromatin relaxation at the *Pparg* promoter and eventually increased PPARγ expression. The results from Ruan et al. demonstrated that more than a third of dynamically expressed lncRNAs were deregulated in a human MASLD cohort and the lncRNA human lncRNA metabolic regulator 1 (hLMR1) positively regulated transcription of genes involved in cholesterol metabolism (*Ruan et al., 2021*). Previous studies have also demonstrated that several lncRNAs, including FLRL2/3/6/7/8, H19, and MALAT-1, were associated with lipogenesis via proteins in the PPAR signaling pathway (*Mukherjee et al., 2022*). Recently, a murine long noncoding single-cell transcriptome analysis elucidated liver lncRNA cell-type specificities, spatial zonation patterns, associated regulatory networks, and temporal patterns of dysregulation during hepatic disease progression. Moreover, a subset of the liver disease-associated regulatory lncRNAs identified have human orthologs (*Karri and Waxman, 2023*). Based on the aforementioned information, lncRNAs emerge as promising candidates for biomarkers and therapeutic targets for MASLD.

Tudor proteins play vital roles in normal cell viability and growth by diverse epigenetic functions, including methylation dependent chromatin-remodeling, histone-binding, pre-RNA-processing, RNA-silencing, and transposon silencing in ligands (*Ying and Chen, 2012*). Tudor proteins are divided into four groups: Group1 Tudor proteins bind to methyl-lysine∕arginine of histone tails, including Tdrd3, PHF1, PHF20, the Jumonji domain-containing protein (JMJD) family and TP53BP1; Group 2 Tudor proteins bind to methyl-arginine of ligands and representative members include SMN and SMNDC1; Group 3 is represented by SND1; and Group 4, contains many Tudor proteins, including Tdrd1-9 and Tdrd11, that have been identified in methylation-dependent association with PIWI proteins Ago3, Aub, and Piwi. In this study, *Snhg3* induced the protein level of SND1 by promoting K63-linked ubiquitination of SND1 and increasing its protein stability. Additionally, *Snhg3*-induced SND1 protein stability seemed subtle, indicating there may be other way for *Snhg3* promotion SND1, such as riboregulation. Some studies suggested that SND1 plays important roles in cancer by interacting with other transcription factors, including PPARγ, signal transducer and activator of transcription 5/6 (STAT6/5) and myeloblastosis oncogene (c-Myb) (*Duan et al., 2014*; *Navarro-Imaz et al., 2020*). SND1 could induce adipogenesis and promote the formation of lipid droplets in adipocytes through working as a co-activator of PPARγ and regulating H3 acetylation (*Duan et al., 2014*). Our study showed that both *Snhg3* and SND1 decreased the H3K27me3 level and promoted the expression of PPARγ. SND1 could interact with *Snhg3* and mediate the *Snhg3*-induced decrease in H3K27me3 and increase in PPARγ expression. Furthermore, inhibition of PPARγ with T0070907 alleviated *Snhg3*- and SND1-induced *Cd36* and *Cidea/c* increase and improved *Snhg3*-aggravated hepatic steatosis. In lncRNA riboregulation, the actions of noncoding RNAs mostly rely on interactions with proteins, including canonical or noncanonical RNA-binding proteins (RBPs). Canonical RBPs, such as heterogeneous nuclear ribonucleoproteins (hnRNPs), polypyrimidine tract binding protein 1 (PTBP1) and human antigen R (HUR), are often involved in posttranscriptional regulation, including pre-mRNA processing, RNA stability, RNA decay, or nuclear export (*Briata and Gherzi, 2020*). Previous studies have demonstrated that some lncRNAs, for example LINC01018, MEG3, APOA4-AS, hLMR1, Blnc1, and LncARSR interact with canonical RBPs to govern the progression of MASLD (*Sommerauer and Kutter, 2022*).

In summary, our study demonstrates that lncRNA-*Snhg3* influenced fatty acid metabolism and aggravated hepatic steatosis under DIO status. Furthermore, *Snhg3* increased the expression, stability, and nuclear localization of SND1 protein by interacting with SND1, thus enhancing the expression of PPARγ via reducing H3K27me3 enrichment and boosting chromatin loose remodeling at the *Pparg* promoter, indicating that SND1/H3K27me3/PPARγ is partially responsible for *Snhg3*-induced hepatic steatosis. This study reveals a new signaling pathway, *Snhg3*/SND1/H3K27me3/PPARγ, responsible for hepatic steatosis and provides evidence of lncRNA-mediated epigenetics in the pathophysiology of MASLD (*Figure 8G*).

However, there are still some limitations to this study that require further investigation. Notably, the expression change of H3K27me3, a global repressive histone mark, may affect multiple downstream target genes, including *Pparg*; therefore, more target genes involved in MASLD need to be elucidated. Moreover, the precise mechanism by which SND1 regulates H3K27me3 is still unclear

and hence requires further investigation. It is crucial to ascertain whether SND1 itself functions as a new demethylase or if it influences other demethylases, such as JMJD3, enhancer of zeste homolog 2 (EZH2), and ubiquitously transcribed tetratricopeptide repeat on chromosome X (UTX). SND1 has multiple roles through associating with different types of RNA molecules, including mRNA, miRNA, circRNA, dsRNA, and lncRNA. SND1 could bind negative-sense SARS-CoV-2 RNA and to promote viral RNA synthesis (*Schmidt et al., 2023*). SND1 is also involved in hypoxia by negatively regulating hypoxia-related miRNAs (*Saarikettu et al., 2023*). Furthermore, a recent study revealed that lncRNA SNAI3-AS1 can competitively bind to SND1 and perturb the m6A-dependent recognition of *Nrf2* mRNA 3'UTR by SND1, thereby reducing the mRNA stability of *Nrf2* (*Zheng et al., 2023*). Huang et al. also reported that circMETTL9 can directly bind to and increase the expression of SND1 in astrocytes, leading to enhanced neuroinflammation (*Huang et al., 2023*). However, whether there is an independent-histone methylation role of SND1/lncRNA-*Snhg3* involved in lipid metabolism in the liver needs to be further investigated.

## Materials and methods
### Animals and treatments

C57BL/6 *Snhg3flox/flox* (*Snhg3*-Flox) mice and hepatocyte-specific *Snhg3* knock-in (*Snhg3*-HKI) mice were created using the CRISPR-Cas9 system at Cyagen Biosciences. To engineer the targeting vector for *Snhg3*-Flox mice, the exon 3 of *Snhg3* was selected as the conditional knockout region, and homology arms and the cKO region were generated by PCR using BAC clone as a template. Cas9 and gRNA were co-injected into fertilized eggs with a targeting vector for mice production. The obtained mice were identified by PCR followed by sequence analysis. Hepatocyte-specific *Snhg3* knock-out (*Snhg3*-HKO) mice were generated by crossing *Snhg3*-Flox mice with C57BL/6-Alb-Cre mice. For *Snhg3*-HKI mice, the 'Alb promoter-mouse *Snhg3* cDNA-polyA' cassette was inserted into an *H11* locus (~0.7 kb 5' of the Eif4enif1 gene and ~4.5 kb 3' of the *Drg1* gene), and homology arms were generated by PCR using BAC clone as template to engineer the targeting vector. Cas9 and gRNA were co-injected into fertilized eggs with targeting vector for mice production. The obtained mice were identified by PCR followed by sequence analysis. All mice were housed in the pathogen-free conditions (SPF) facility and maintained on a 12 hr light-dark cycle and a regular unrestricted diet. All mice were fed either a normal chow diet (9% fat; Lab Diet) or HFD (60% fat, Research Diets) for inducing obesity and libitum with free access to water. Unless otherwise noted, 6~8-week-old male mice were used for all experiments. 8-week-old mice fed on HFD were injected intraperitoneally (i.p.) with 1 mg/kg of T0070907 dissolved in DMSO for 5 days per week for 2 months. Liver tissue samples were analyzed by the High Fatty Sample Total Cholesterol (TC) Content Assay Kit (APPLYGEN, Cat#E1025-105) and the High Fatty Sample Triglyceride (TG) Content Assay Kit (APPLYGEN, Cat#E1026-105), respectively. Serum concentrations of ALT, AST, FFAs, TG and TC were determined using an automated Monarch device (Peking Union Medical College Hospital, Beijing, China). Serum insulin was detected using a mouse insulin ELISA kit (JM-02862M1, Beijing BioDee Biotechnology Co., Ltd.). All animal experiments were conducted under protocols approved by the Animal Research Committee of the Institute of Laboratory Animals, Institute of Basic Medical Sciences Chinese Academy of Medical Sciences & School of Basic Medicine Peking Union Medical College (ACUC-A01-2022-010).

### Cell culture

Primary mouse hepatocytes were isolated from 8-week-old male C57BL/6 J mice and cultured in RPMI 1640 medium with 10% FBS as previously described (*Matsumoto et al., 2002*). Hepa1-6 cells (ATCC, Cat#CRL-1830) were cultured at 37 °C, 5% $CO_2$ in DMEM (Gibco, Carlsbad, USA) supplemented with 10% FBS, and 1% penicillin-streptomycin. The species origin of Hepa1-6 cell was confirmed with PCR and the identity of Hepa1-6 cell was authenticated with STR profiling. Hepa1-6 cell line was checked negative for mycoplasma by PCR. After attachment, the cells were transfected with indicated plasmids or *siSnd1* by Lipofectamine 3000 Transfection Kit (Invitrogen). Cells were treated with and without 0.25 mM PA for 12h - 24h before collection. The sequences of *siSnd1* were seen key resources table.

## Plasmid construction

*Snhg3* was amplified from liver cDNA and was then constructed into pcDNA3.1 using Kpn I and EcoR I. The primers were seen key resources table.

## Real-time quantitative PCR (RT-qPCR)

Total RNA was extracted from mouse tissues using a Trizol-based method. Approximately 2 μg of total RNA was reverse-transcribed into a first-strand cDNA pool using reverse transcriptase and random primers, according to the manufacturer's instructions. RT-qPCR was performed using SYBR Green PCR Master Mix (A6002, Promega) with the gene-specific primers (key resources table). All gene expression data were normalized to *β-Actin* expression levels.

## Western blotting

Protein was extracted from frozen tissue samples in cell lysis buffer. In total, protein was loaded onto a 10% SDS-polyacrylamide gel, and separated proteins were transferred to PVDF membranes. Western blot assays were performed using indicated specific antibodies (key resources table). The proteins were quantified by ImageJ software.

## Coding potential prediction

The coding potential of *Snhg3* was evaluated by the Coding Potential Calculator at CPC2@CBI,PKU( gao-lab.org) (*Kang et al., 2017*).

## Histopathologic analysis

Liver tissue sections were fixed in 4% paraformaldehyde, then embedded in paraffin and stained with H&E to visualize the general morphological and structural characteristics of tissues. Lipid droplet accumulation in the liver was visualized using Oil red O staining of frozen liver sections that were prepared in optimum cutting temperature (O.C.T.) compound. Liver fibrosis was visualized using Picro Sirius Red Stain.

## Subcellular fractionation

A Cytoplasmic & Nuclear fraction Kit (Beyotime Biotechnology, China) was used to detect *Snhg3* expression in cytoplasmic and nuclear fractions. RNA was extracted from the cytoplasmic and nuclear fractions using a Trizol-based method and subjected to qPCR. *Gapdh* was used as a cytoplasmic marker, and *Neat1* and *Xist* were used as a nuclear marker. The percentage of the transcript abundance was calculated using the following formula, Nucleus $\% = 2^{\wedge Ct(Nucleus)}/(2^{\wedge Ct(Cytoplasm)} + 2^{\wedge Ct(Nucleus)})$, Cytoplasm $\% = 1 -$ Nucleus %.

## Mouse calorimetry

Male mice were housed individually in metabolic chambers of an Oxymax system (Columbus Instruments). The first readings were taken after a 24 hr acclimation period. Heat production, total carbon dioxide production and oxygen consumption, and RER were determined by Comprehensive laboratory animal monitoring system (CLAMS). The data were analyzed with CalR (*Mina et al., 2018*).

## Insulin tolerance test (ITT) and glucose tolerance test (GTT)

For ITT, male mice fasted for 6 hr and received an intraperitoneal injection of human insulin (0.75 IU/ kg). For GTT, male mice fasted for 6 hr or 16 hr received an intraperitoneal injection of glucose (1 g/ kg). A series of blood glucose concentrations were measured from tail blood at the indicated times using a One-Touch Ultra glucometer (LifeScan Inc, Milpitas, CA).

## Chromatin Immunoprecipitation (ChIP)

The ChIP assay was performed using Sonication ChIP Kit (Abclonal, Cat#RK20258). Briefly, the liver tissues or primary hepatocytes were collected and cross-linking fixed. Cross-linked chromatin fragments were precipitated with Rabbit control IgG (Abclonal, Cat#AC005) or anti-H3K27me3 antibody (Abclonal, Cat# A16199) for subsequent PCR analysis using the amplification primers for mouse *Pparg* promoter (+101 ~+ 420 bp; key resources table).

## RNA immunoprecipitation (RIP)

The RIP assay was performed using RIP Assay Kit (BersinBio, Cat#Bes5101). Briefly, Hepa1-6 cells were transfected by indicated plasmids, respectively. The cells were collected, cross-linking fixed and precipitated with Mouse Control IgG (Abclonal, Cat#AC011) or anti-FLAG antibody (Abclonal, Cat#AE005) for subsequent RT-qPCR analysis using the amplification primers for *Snhg3*.

## RNA sequencing (RNA-Seq)

The RNA-Seq was performed according to the manufacturer's protocol (BGI-Shenzhen, https://www.yuque.com/yangyulan-ayaeq/oupzan/fuoao4). Briefly, total RNA was extracted from liver of three male DIO (27 weeks) mice and three male control mice for RNA-Seq to screen the differentially expressed lncRNAs using Trizol (Invitrogen, Carlsbad, CA, USA) according to manual instruction. rRNA in total RNA removed using RNase H kit, was subsequently to construct library and perform sequencing analysis. The data were mapped to mouse genome (GRCm39) by using Bowtie2. The data for the differentially expressed lncRNAs had been deposited to National Genomics Data Center, China National Center for Bioinformation (NGDC-CNCB) (https://ngdc.cncb.ac.cn/) with the dataset identifier CRA009822. Additionally, total RNA was extracted from livers of three male DIO *Snhg3*-HKI mice and three male DIO WT mice for RNA-Seq to screen the differentially expressed mRNAs using Trizol. Total RNA was enriched by oligo (dT)-attached magnetic beads, followed by library construction and sequencing analysis. The data were mapped to mouse genome (GRCm39) by using Bowtie2. The data for the differentially expressed mRNAs have been deposited to the Sequence Read Archive (SRA) (submit.ncbi.nlm.nih.gov) with the dataset identifier SRR22368163, SRR22368164, SRR22368165, SRR22368166, SRR22368167, and SRR22368168.

## Assay for transposase-accessible chromatin with high-throughput sequencing (ATAC-Seq)

The ATAC-seq was performed according to manufacturer's protocols (BGI_Shenzhen, https://www.yuque.com/yangyulan-ayaeq/oupzan/lllmzg). Briefly, fresh liver tissue samples from three male DIO *Snhg3*-HKI mice and three male DIO control mice were flash frozen by liquid nitrogen and then ground completely. The transposition reactions were initiated by adding transposase. The PCR reaction system was configured to initiate PCR amplification of the transposition products. The corresponding library quality control protocol would be selected depending on product requirements. Single-stranded PCR products were produced via denaturation. The reaction system and program for circularization were subsequently configured and set up. Single-stranded cyclized products were produced, while uncyclized linear DNA molecules were digested. Single-stranded circle DNA molecules were replicated via rolling cycle amplification, and a DNA nanoball (DNB) which contain multiple copies of DNA was generated. Sufficient quality DNBs were then loaded into patterned nanoarrays using high-intensity DNA nanochip technique and sequenced through combinatorial Probe-Anchor Synthesis (cPAS). Data were filtered by removing adaptor sequences, contamination and low-quality reads from raw reads. Bowtie2 was used to do genome alignment after evaluating its performance.

Peak Calling was performed by MACS (Model-based Analysis for ChIP-Seq). The candidate Peak region was extended to be long enough for modeling. Dynamic Possion Distribution was used to calculate the p-value of the specific region based on the unique mapped reads. The region would be defined as a Peak when the p-value <1e-05. MACS works well for the identification of the sharp peaks of most sequence-specific transcription factors.

Peak Distribution on Gene Elements. Peaks were classified based on the location (UCSC annotation data) and showed in the following genome regions: promoter (≤1 kb), promoter (1–2 kb), promoter (2–3 kb), 5'-UTR, 3'-UTR, intergenic, introns, and exons.

Differential Peaks were identified using MAnorm. First, the true intensities of the most common peaks were assumed to be the same between two ATAC-Seq samples. Second, the observed differences in sequence read density in common peaks were presumed to reflect the scaling relationship of ATAC-Seq signals between two samples, which could thus be applied to all peaks. Based on these hypotheses, the log2 ratio of read density between two samples M was plotted against the average log2 read density A for all peaks, and robust linear regression was applied to fit the global dependence between the M-A values of common peaks. Then the derived linear model was used as a reference for normalization and extrapolated to all peaks. Finally, the p-value for each Peak was calculated based on

the Bayesian model, the significant regions were picked up if |M=log2FC|>2 and p-value <0.001. To identify DARs, the count matrix was input into MAnorm with the cutoff of abs(log2FC)>2 and p-value <0.001. Genomic features of DARs were annotated by R package ChIPseeker (v1.30.3).

Differential motifs analysis. After extracting the corresponding peak sequence, Hypergeometric Optimization of Motif EnRichment (HOMER, (ucsd.edu)) and Cis-Regulatory Element Motif Activities (CREMA (unibas.ch)) were used for motif analysis. Genomic regions with differential ATAC peaks were shown using IGV software.

The data of ATAC-seq has been deposited to National Genomics Data Center, China National Center for Bioinformation (NGDC-CNCB) (https://ngdc.cncb.ac.cn/) with the dataset identifier CRA009511.

## Integrated analysis ATAC-Seq data with RNA-Seq data

The common number and unique number of genes associated with DARs and DEGs were counted using Wayne comparative analysis. The values of each quadrant satisfying the log2FC condition were selected from DARs-associated genes and DEGs to draw a nine-quadrant plot, respectively. Pearson correlation was calculated for both sets of data and p-value was calculated using the Z-test.

## Cleavage Under Targets and Tagmentation sequencing (CUT&Tag-Seq)

CUT&Tag was performed according to the Hyperactive Universal CUT&Tag Assay Kit for Illumina (Vazyme, Nanjing, Cat#TD903-01). Briefly, the mixed liver tissues from three DIO *Snhg3*-HKO mice were used for the CUT&Tag experiment. pA-Tn5 transposase was used to cut the genome and add a special adaptor sequence to build a library. The single-stranded PCR products were sequenced on illumina/DNBSEQ-T7 platform PE150 (Annoroad Gene Technology Co.Itd). The data were mapped to mouse genome (GRCm38) by using Bowtie2. Genomic features of DARs were annotated by R package ChIPseeker (v1.30.3). The data of CUT&Tag has been deposited to National Genomics Data Center, China National Center for Bioinformation (NGDC-CNCB) (https://ngdc.cncb.ac.cn/) with the dataset identifier CRA009582.

## Biotin-RNA pull-down and mass spectrometry assay

Biotin-RNA pull-down assay was performed as described in a previous study (*Guo et al., 2018*). Briefly, *Snhg3* DNA was amplified from mouse liver cDNA using the primers listed in the key resources table and lacZ DNA fragment were constructed into pGEM-T easy vector. The pGEM-T-*Snhg3* and pGEM-T-lacZ vectors were linearized by restriction enzyme digestion, then transcribed to *Snhg3* and lacZ fragments. Biotinylated RNAs were transcribed in vitro with Biotin-RNA Labeling Mix (Roche, Indianapolis, IN) and purified with quick spin RNA columns (Roche, Indianapolis, IN). Biotin-labeled RNA or unbiotinylated RNAs was dissolved in RNA structure buffer (10 mM Tris, pH 7.0, 0.1 M KCl, 10 mM MgCl2) to allow formation of the secondary structure. Primary hepatocytes lysates were added to biotin-labeled RNA or unbiotinylated RNA. Streptavidin agarose beads (GE Healthcare, Little Chalfont, UK) were mixed with a pull-down reaction and then rotated constantly. RNA affinity captures were subjected to 12% SDS-PAGE followed by coomassie blue staining or Western blotting. The various bands that were visualized by coomassie blue staining were excised and subjected to mass spectrometry analyses (LC-MS/MS, A TripleTOF , ABsciex, Concord, ON). The data of RNA pull-down for *Snhg3*, control or lacZ were deposited to the iProX (https://www.iprox.cn/) with the dataset identifier PXD039526.

## Ubiquitination assays

For endogenous ubiquitination assays, Hepa1-6 cells were transfected with the indicated combinations of plasmids, including HA-ubiquitin and *Snhg3* plasmids. For exogenous ubiquitination assays, Hepa1-6 cells were transfected with the indicated combinations of plasmids, including HA-ubiquitin, HA-K33-ubiquitin, HA-K63-ubiquitin, HA-K48-ubiquitin, Flag-SND1 and *Snhg3* plasmids. Cells were treated with 20 μM MG132 proteasome inhibitor (M1902, AbMole) for 6 hr prior to lyse in lysis buffer 200 mM NaCl, 20 mM Tris-HCl (pH 7.4), 2.5 mM MgCl2, 0.5% Triton X-100, 1 mM PMSF, and protease inhibitor cocktail and then were sonicated. After centrifugation at 14,000 × *g*, the cleared lysates were subjected to immunoprecipitation with anti-SND1 antibody (sc-166676, Santa Cruz) for endogenous ubiquitination assays or with anti-DDDDK-tag magnetic beads (M185-10R, MBL) for exogenous

ubiquitination assays. The immunocomplexes were collected and subjected to estern blotting with the indicated antibodies.

## Statistical analysis

Data analyses were performed with SPSS (Version 17.0, SPSS, Inc). The curves of body weight and ITT were analyzed using a repeated measure two-way ANOVA. For the other statistical analysis, the Kolmogorov-Smirnov test was firstly used for normality test. For the data conforming to the normal distribution, experiments were analyzed using Independent-Samples T-test or one-way ANOVA. All data were presented as the mean ± SD or the mean ± SEM. * $p < 0.05$ was considered statistically significant.

## Acknowledgements

This work was supported by Chinese Academy of Medical Sciences Innovation Fund for Medical Sciences (CIFMS2021-I2M-1–016 to XL), National Key R&D Program of China (2022YFC2504002 to LY, 2022YFC2504003 to XL), Beijing Natural Science Foundation (7242094 to XL), the National Natural Science Foundation of China (82270925 to AQ), High-level New R&D Institute of Department of Science and Technology of Guangdong Province (2019B090904008 to AQ), and High-level Innovative Research Institute of Department of Science and Technology of Guangdong Province (2021B0909050003 to AQ). We thank Dr. Yi Li (Cancer Hospital, Chinese Academy of Medical Sciences and Peking Union Medical College) for providing the plasmid of HA-Ub, HA-Ub-K63 and HA-Ub-K48. We thank Cyagen Biosciences for the collaborative efforts in the creation of *Snhg3*-loxP and liver-specific knock-in *Snhg3* mice. We thank BGI-Shenzhen for the collaborative efforts in RNA-Seq and ATAC-Seq technology. We thank Anoroad gene technology (Beijing) for the collaborative efforts in CUT&Tag-Seq and Jingjie PTM BioLab (Hangzhou) Co. Ltd for the collaborative efforts in mass spectrometry technology resources.

## Additional information

### Funding

| Funder | Grant reference number | Author |
| --- | --- | --- |
| Chinese Academy of Medical Sciences Innovation Fund for Medical Sciences | CIFMS2021-I2M-1-016 | Xiaojun Liu |
| Beijing Natural Science Foundation | 7242094 | Xiaojun Liu |
| National Key R&D Program of China | 2022YFC2504002 | Li Yan |
| National Key R&D Program of China | 2022YFC2504003 | Xiaojun Liu |
| the National Natural Science Foundation of China | 82270925 | Aijun Qiao |
| High-level New R&D Institute of Department of Science and Technology of Guangdong Province | 2019B090904008 | Aijun Qiao |
| High-level Innovative Research Institute of Department of Science and Technology of Guangdong Province | 2021B0909050003 | Aijun Qiao |

The funders had no role in study design, data collection and interpretation, or the decision to submit the work for publication.

## Author contributions
Xianghong Xie, Data curation, Formal analysis, Investigation, Writing - original draft; Mingyue Gao, Wei Zhao, Chunmei Li, Weihong Zhang, Jiahui Yang, Yinliang Zhang, Enhui Chen, Yanfang Guo, Zeyu Guo, Minglong Zhang, Yinghan Zhu, Yiting Wang, Investigation; Ebenezeri Erasto Ngowi, Writing – review and editing; Heping Wang, Xiaoman Wang, Formal analysis; Xiaolu Li, Methodology; Hong Yao, Fude Fang, Resources, Writing – review and editing; Li Yan, Resources, Funding acquisition, Writing – review and editing; Meixia Li, Supervision, Project administration, Writing – review and editing; Aijun Qiao, Xiaojun Liu, Supervision, Funding acquisition, Project administration, Writing – review and editing

## Author ORCIDs
Xianghong Xie (ID) http://orcid.org/0009-0001-3481-469X
Heping Wang (ID) https://orcid.org/0000-0002-0153-478X
Meixia Li (ID) https://orcid.org/0000-0001-9992-3275
Aijun Qiao (ID) http://orcid.org/0000-0001-7545-3395
Xiaojun Liu (ID) https://orcid.org/0000-0003-4672-8569

## Ethics
All animal experiments were conducted under protocols approved by the Animal Research Committee of the Institute of Laboratory Animals, Institute of Basic Medical Sciences Chinese Academy of Medical Sciences & School of Basic Medicine Peking Union Medical College (ACUC-A01-2022-010).

Reviewer #1 (Public Review): https://doi.org/10.7554/eLife.96988.4.sa1
Reviewer #2 (Public Review): https://doi.org/10.7554/eLife.96988.4.sa2
Author response https://doi.org/10.7554/eLife.96988.4.sa3

---

# Additional files

## Supplementary files
• MDAR checklist

## Data availability
The lncRNA-Seq data had been deposited to National Genomics Data Center, China National Center for Bioinformation (NGDC-CNCB) (https://ngdc.cncb.ac.cn/) with the dataset identifier CRA009822. The data of RNA pull-down for Snhg3, control or lacZ have been deposited to the iProX (https://www.iprox.cn/) with the dataset identifier IPX0005781000 (ProteomeXchange identifier: PXD039526). The data of RNA-seq have been deposited to the Sequence Read Archive (SRA) with the dataset identifier SRR22368163, SRR22368164, SRR22368165, SRR22368166, SRR22368167 and SRR22368168. The data of ATAC-seq and Cut&Tag have been deposited to National Genomics Data Center, China National Center for Bioinformation (NGDC-CNCB) (https://ngdc.cncb.ac.cn/) with the dataset identifier CRA009511 and CRA009582, respectively.

The following datasets were generated:

| Author(s) | Year | Dataset title | Dataset URL | Database and Identifier |
|---|---|---|---|---|
| Xie X, Gao M, Wang H, Zhang M, Zhao W, Li C, Zhang W, Yang J, Zhang Y, Chen E, Guo Y, Guo Z, Ngowi EE, Wang X, Zhu Y, Wang Y, Li X, Yao H, Yan L, Fang F, Li M, Qiao A, Liu X | 2024 | The lncRNA-seq of mice liver | https://ngdc.cncb.ac.cn/gsa/search?searchTerm=CRA009822 | Genome Sequence Archive, CRA009822 |

*Continued on next page*

Continued

| Author(s) | Year | Dataset title | Dataset URL | Database and Identifier |
|---|---|---|---|---|
| Xie X, Gao M, Wang H, Zhang M, Zhao W, Li C, Zhang W, Yang J, Zhang Y, Chen E, Guo Y, Guo Z, Ngowi EE, Wang X, Zhu Y, Wang Y, Li X, Yao H, Yan L, Fang F, Li M, Qiao A, Liu X | 2024 | RNA-Pull down and MS for Snhg3 | https://www.iprox.cn//page/project.html?id=IPX0005781000 | Integrated Proteome Resources, IPX0005781000 |
| Xie X, Gao M, Wang H, Zhang M, Zhao W, Li C, Zhang W, Yang J, Zhang Y, Chen E, Guo Y, Guo Z, Ngowi EE, Wang X, Zhu Y, Wang Y, Li X, Yao H, Yan L, Fang F, Li M, Qiao A, Liu X | 2024 | The ATAC-seq of diet induced obesity Snhg3-LKI mice liver | https://ngdc.cncb.ac.cn/gsa/search?searchTerm=CRA009511 | Genome Sequence Archive, CRA009511 |
| Xie X, Gao M, Wang H, Zhang M, Zhao W, Li C, Zhang W, Yang J, Zhang Y, Chen E, Guo Y, Guo Z, Ngowi EE, Wang X, Zhu Y, Wang Y, Li X, Yao H, Yan L, Fang F, Li M, Qiao A, Liu X | 2024 | The CUT&TAG-seq of Snhg3-LKI mice liver | https://ngdc.cncb.ac.cn/gsa/search?searchTerm=CRA009582 | Genome Sequence Archive, CRA009582 |
| Xie X, Gao M, Wang H, Zhang M, Zhao W, Li C, Zhang W, Yang J, Zhang Y, Chen E, Guo Y, Guo Z, Ngowi EE, Wang X, Zhu Y, Wang Y, Li X, Yao H, Yan L, Fang F, Li M, Qiao A, Liu X | 2024 | lncRNA_Snhg3 regulates hepatic lipid metabolism | https://www.ncbi.nlm.nih.gov/sra/?term=SRR22368163 | NCBI Sequence Read Archive, SRR22368163 |
| Xie X, Gao M, Wang H, Zhang M, Zhao W, Li C, Zhang W, Yang J, Zhang Y, Chen E, Guo Y, Guo Z, Ngowi EE, Wang X, Zhu Y, Wang Y, Li X, Yao H, Yan L, Fang F, Li M, Qiao A, Liu X | 2024 | lncRNA_Snhg3 regulates hepatic lipid metabolism | https://www.ncbi.nlm.nih.gov/sra/?term=SRR22368164 | NCBI Sequence Read Archive, SRR22368164 |
| Xie X, Gao M, Wang H, Zhang M, Zhao W, Li C, Zhang W, Yang J, Zhang Y, Chen E, Guo Y, Guo Z, Ngowi EE, Wang X, Zhu Y, Wang Y, Li X, Yao H, Yan L, Fang F, Li M, Qiao A, Liu X | 2024 | lncRNA_Snhg3 regulates hepatic lipid metabolism | https://www.ncbi.nlm.nih.gov/sra/?term=SRR22368165 | NCBI Sequence Read Archive, SRR22368165 |
| Xie X, Gao M, Wang H, Zhang M, Zhao W, Li C, Zhang W, Yang J, Zhang Y, Chen E, Guo Y, Guo Z, Ngowi EE, Wang X, Zhu Y, Wang Y, Li X, Yao H, Yan L, Fang F, Li M, Qiao A, Liu X | 2024 | lncRNA_Snhg3 regulates hepatic lipid metabolism | https://www.ncbi.nlm.nih.gov/sra/?term=SRR22368166 | NCBI Sequence Read Archive, SRR22368166 |

*Continued*

| Author(s) | Year | Dataset title | Dataset URL | Database and Identifier |
|---|---|---|---|---|
| Xie X, Gao M, Wang H, Zhang M, Zhao W, Li C, Zhang W, Yang J, Zhang Y, Chen E, Guo Y, Guo Z, Ngowi EE, Wang X, Zhu Y, Wang Y, Li X, Yao H, Yan L, Fang F, Li M, Qiao A, Liu X | 2024 | lncRNA_Snhg3 regulates hepatic lipid metabolism | https://www.ncbi.nlm.nih.gov/sra/?term=SRR22368167 | NCBI Sequence Read Archive, SRR22368167 |
| Xie X, Gao M, Wang H, Zhang M, Zhao W, Li C, Zhang W, Yang J, Zhang Y, Chen E, Guo Y, Guo Z, Ngowi EE, Wang X, Zhu Y, Wang Y, Li X, Yao H, Yan L, Fang F, Li M, Qiao A, Liu X | 2024 | lncRNA_Snhg3 regulates hepatic lipid metabolism | https://www.ncbi.nlm.nih.gov/sra/?term=SRR22368168 | NCBI Sequence Read Archive, SRR22368168 |
| Xie X, Gao M, Wang H, Zhang M, Zhao W, Li C, Zhang W, Yang J, Zhang Y, Chen E, Guo Y, Guo Z, Ngowi EE, Wang X, Zhu Y, Wang Y, Li X, Yao H, Yan L, Fang F, Li M, Qiao A, Liu X | 2024 | RNA-Pull down and MS for Snhg3 | https://proteomecentral.proteomexchange.org/cgi/GetDataset?ID=PXD039526 | ProteomeXchange, PXD039526 |

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

# Appendix 1

## Appendix 1—key resources table

| Reagent type (species) or resource | Designation | Source or reference | Identifiers | Additional information |
|---|---|---|---|---|
| Gene (*M. musculus*) | LncRNA *Snhg3* | GenBank | NR_003270.2 | |
| Gene (*M. musculus*) | *Snd1* | GenBank | NM_019776.2 | |
| Strain, strain background (*Escherichia coli*) | Trans5α | TransGen Biotech | Cat#CD201 | |
| Strain, strain background (*M. musculus*) | *Snhg3*<sup>flox/flox</sup> mice | This paper | | *Snhg3*<sup>flox/flox</sup> mice were created using the CRISPR-Cas9 system at Cyagen Biosciences. |
| Strain, strain background (*M. musculus*) | Hepatocyte-specific knock-in *Snhg3* (*Snhg3*-HKI) mice | This paper | | *Snhg3*-HKI mice were created using the CRISPR-Cas9 system at Cyagen Biosciences. |
| Strain, strain background (*M. musculus*) | Alb-Cre transgenic mice | Cyagen | Cat#C001006 | |
| Strain, strain background (*M. musculus*) | C57BL/6 (*Wild type*) | HFK BIOSCIENCE | | male |
| Genetic reagent (*Homo-sapiens*) | Ad-*SND1* (adenovirus) | WZ Biosciences | Cat#VH832073 | Adenovirus infect cells to express SND1 (*human*) |
| Genetic reagent (*M. musculus*) | Ad-*Snhg3* (adenovirus) | This paper | N/A | Adenovirus infect cells to express *Snhg3 (mouse)* |
| Cell line (*M. musculus*) | Hepa 1–6 | ATCC | Cat#CRL-1830 | |
| Biological sample (*M. musculus*) | Primary hepatocytes | This paper | | Freshly isolated from C57BL/6 |
| Antibody | anti-GAPDH (Rabbit polyclonal) | CWBio | Cat#CW0100M; RRID: AB_2801390 | WB (1:1000) |
| Antibody | anti-β-Actin (Rabbit polyclonal) | Abclonal | Cat#AC026; RRID: AB_2768234 | WB (1:50000) |
| Antibody | anti-H3 (Rabbit polyclonal) | Abclonal | Cat#A17562; RRID: AB_2770395 | WB (1:1000) |
| Antibody | anti-H3K27me3 (Rabbit polyclonal) | Abclonal | Cat# A16199; RRID: AB_2763651 | WB (1:1000) IP (1:100) |
| Antibody | anti-SND1 (Rabbit polyclonal) | Abclonal | Cat#A5874; RRID: AB_2766623 | WB (1:1000) |
| Antibody | anti-SND1 (Mouse monoclonal) | Santa Cruz biotechnology | Cat#sc-166676; RRID: AB_2270808 | WB (1:500) IP (1:50) |
| Antibody | anti-PPARγ (Rabbit polyclonal) | Abclonal | Cat#A11183; RRID: AB_2758449 | WB (1:500) |
| Antibody | anti-CD36 (Rabbit polyclonal) | Abclonal | Cat#A14714; RRID: AB_2761590 | WB (1:1000) |
| Antibody | anti DDDDK-Tag (Mouse monoclonal) | Abclonal | Cat#AE005; RRID: AB_2770401 | WB (1:1000) IP (1:100) |
| Antibody | anti HA-Tag (Rabbit polyclonal) | Abclonal | Cat#AE036; RRID: AB_2771924 | WB (1:1000) |
| Antibody | anti-Ub (Rabbit polyclonal) | Abclonal | Cat#A19686; RRID: AB_2862735 | WB (1:1000) |
| Antibody | anti-Ub (K33) (Rabbit polyclonal) | Abclonal | Cat# A18199; RRID: AB_2861976 | WB (1:1000) |
| Antibody | anti-Ub (K48) (Rabbit polyclonal) | Abclonal | Cat#A18163; RRID: AB_2861948 | WB (1:1000) |
| Antibody | anti-Ub (K63) (Rabbit polyclonal) | Abclonal | Cat# A18164; RRID: AB_2861949 | WB (1:1000) |

*Appendix 1 Continued on next page*

*Appendix 1 Continued*

| Reagent type (species) or resource | Designation | Source or reference | Identifiers | Additional information |
|---|---|---|---|---|
| Antibody | Mouse Control IgG (Rabbit polyclonal) | Abclonal | Cat#AC011; RRID: AB_2770414 | WB (1:1000) IP (1:100) |
| Antibody | Rabbit Control IgG (Rabbit polyclonal) | Abclonal | Cat#AC005; RRID: AB_2771930 | WB (1:1000) IP (1:100) |
| Antibody | Goat anti-mouse IgG (H+L) (Rabbit polyclonal) | ZSGB-Bio | Cat#ZB-2305; RRID: AB_2747415 | WB (1:10000) |
| Antibody | Goat anti-rabbit IgG (H+L) (Rabbit polyclonal) | ZSGB-Bio | Cat#ZB-2306; RRID: AB_2868454 | WB (1:10000) |
| Recombinant DNA reagent | pcDNA3.1-*mSnhg3* (Plasmid) | This paper | | Plasmid construct to transfect and express the *Snhg3* |
| Recombinant DNA reagent | PGEM-Teasy-*mSnhg3* (Plasmid) | This paper | | Plasmid construct to cloning and amplification the *Snhg3* |
| Recombinant DNA reagent | pCMV3-Flag-mSND1 (Plasmid) | Sino Biological | Cat#MG52839-NF | Plasmid construct to transfect and express the SND1 |
| Recombinant DNA reagent | HA-Ub (Plasmid) | This paper | | Plasmid construct to transfect and express the HA-UB |
| Recombinant DNA reagent | HA-Ub (K48O) (Plasmid) | This paper | | Plasmid construct to transfect and express the HA-UB (K48O) |
| Recombinant DNA reagent | HA-Ub (K63O) (Plasmid) | This paper | | Plasmid construct to transfect and express the HA-UB (K63O) |
| Sequence-based reagent | *Snhg3*-F | This paper | verexpressing and adenoviral plasmid construction | ATATCGGGTACCGACTTCCGGGCGTTAC |
| Sequence-based reagent | *Snhg3*-R | This paper | verexpressing and adenoviral plasmid construction | ATGATCGAATTCAGACATTCAAATGCT |
| Sequence-based reagent | *Snhg3*-HKO-F | This paper | sgRNA target sequences for knockout mice construction | GTCGAATGGATGAGTTATGTGGG |
| Sequence-based reagent | *Snhg3*-HKO-R | This paper | sgRNA target sequences for knockout mice construction | GATATCCACGTTGGAATGTCTGG |
| Sequence-based reagent | *Snhg3*-HKO (mouse)-F | This paper | Primers for genotyping the transgenic mice | TCTGGAGTGTGAGATAGGAAACTG |
| Sequence-based reagent | *Snhg3*-HKO (mouse)-R | This paper | Primers for genotyping the transgenic mice | TCACTGAGGGTCTTAACTTTTCCAT |
| Sequence-based reagent | *Snhg3*-HKI (mouse)-F1 | This paper | Primers for genotyping the transgenic mice | CTCTACTGGAGGAGGACAAACTG |
| Sequence-based reagent | *Snhg3*-HKI (mouse)-F2 | This paper | Primers for genotyping the transgenic mice | GCATCTGACTTCTGGCTAATAAAG |
| Sequence-based reagent | *Snhg3*-HKI (mouse)-R | This paper | Primers for genotyping the transgenic mice | GTCTTCCACCTTTCTTCAGTTAGC |
| Sequence-based reagent | Alb-cre (mouse)-F1 | This paper | Primers for genotyping the transgenic mice | TGCAAACATCACATGCACAC |
| Sequence-based reagent | Alb-cre (mouse)-F2 | This paper | Primers for genotyping the transgenic mice | GAAGCAGAAGCTTAGGAAGATGG |
| Sequence-based reagent | Alb-cre (mouse)-R | This paper | Primers for genotyping the transgenic mice | TTGGCCCCTTACCATAACTG |
| Sequence-based reagent | si*Snd1*#1F | This paper | siRNA target sequences for knockdown cells construction | GAGAACAUGCGCAAUGACATT |
| Sequence-based reagent | si*Snd1*#1R | This paper | siRNA target sequences for knockdown cells construction | UGUCAUUGCGCAUGUUCUCTT |
| Sequence-based reagent | si*Snd1*#2F | This paper | siRNA target sequences for knockdown cells construction | GCAUGUCUUCUACAUCGACTT |

*Appendix 1 Continued on next page*

Appendix 1 Continued

| Reagent type (species) or resource | Designation | Source or reference | Identifiers | Additional information |
|---|---|---|---|---|
| Sequence-based reagent | si*Snd1*#2R | This paper | siRNA target sequences for knockdown cells construction | GUCGAUGUAGAAGACAUGCTT |
| Sequence-based reagent | si*Snd1*#3F | This paper | siRNA target sequences for knockdown cells construction | GUAUUGCCAGCUCAAGCCA CAGAGUAUTT |
| Sequence-based reagent | si*Snd1*#3R | This paper | siRNA target sequences for knockdown cells construction | AUACUCUGUGGCUUGAGCU GGCAAUACTT |
| Sequence-based reagent | si*Control*-F | This paper | siRNA target sequences for knockdown cells construction | UUCUCCGAACGUGUCACGUTT |
| Sequence-based reagent | si*Control*-R | This paper | siRNA target sequences for knockdown cells construction | ACGUGACACGUUCGGAGAATT |
| Sequence-based reagent | promoter region (+101 ~+ 420bp)-F | This paper | Primers of *Pparγ* promoter segment for ChIP-qPCR assay | TATTGGGTCGCGCGCAGCC |
| Sequence-based reagent | promoter region (+101 ~+ 420bp)-R | This paper | Primers of *Pparγ* promoter segment for ChIP-qPCR assay | ACACAGTCCTGTCAGAACG |
| Sequence-based reagent | Mouse *β-Actin*-F | This paper | Primers for qPCR | CCAGCCTTCCTTCTTGGGTAT |
| Sequence-based reagent | Mouse *β-Actin*-R | This paper | Primers for qPCR | TGCTGGAAGGTGGACAGTGAG |
| Sequence-based reagent | Mouse *Gapdh*-F | This paper | Primers for qPCR | GGAGAGTGTTTCCTCGTCCC |
| Sequence-based reagent | Mouse *Gapdh*-R | This paper | Primers for qPCR | ATGAAGGGGTCGTTGATGGC |
| Sequence-based reagent | Mouse *Xist*-F | This paper | Primers for qPCR | AGACTACAGGATGAATTTGGAGTC |
| Sequence-based reagent | Mouse *Xist*-R | This paper | Primers for qPCR | ATTGTTTGTCCCTTTGGGCTC |
| Sequence-based reagent | Mouse *Neat1*-F | This paper | Primers for qPCR | AGGAGTTAGTGACAAGGAGG |
| Sequence-based reagent | Mouse *Neat1*-R | This paper | Primers for qPCR | TGCCTTCCACACGTCCACTG |
| Sequence-based reagent | Mouse *Snhg3*-F | This paper | Primers for qPCR | CTCTCTAGGCGTCGCTCTCT |
| Sequence-based reagent | Mouse *Snhg3*-R | This paper | Primers for qPCR | CTTCTAATGGCCGAGGCTGT |
| Sequence-based reagent | Mouse *Snd1*-F | This paper | Primers for qPCR | CACCCTGACACTTCCAGTCC |
| Sequence-based reagent | Mouse *Snd1*-R | This paper | Primers for qPCR | ACAATTATGGCGCACCCAGA |
| Sequence-based reagent | Mouse *Pparγ*-F | This paper | Primers for qPCR | TCAGCTCTGTGGACCTCTCC |
| Sequence-based reagent | Mouse *Pparγ*-R | This paper | Primers for qPCR | ACCCCTTGCATCCTTCACAAG |
| Sequence-based reagent | Mouse *Cd36*-F | This paper | Primers for qPCR | GGAGCAACTGGTGGATGGTT |
| Sequence-based reagent | Mouse *Cd36*-R | This paper | Primers for qPCR | CTACGTGGCCCGGTTCTAAT |
| Sequence-based reagent | Mouse *Cidea*-F | This paper | Primers for qPCR | AGGCCGTGTTAAGGAATCTG |
| Sequence-based reagent | Mouse *Cidea*-R | This paper | Primers for qPCR | AACCAGCCTTTGGTGCTAGG |

Appendix 1 Continued on next page

*Appendix 1 Continued*

| Reagent type (species) or resource | Designation | Source or reference | Identifiers | Additional information |
|---|---|---|---|---|
| Sequence-based reagent | Mouse *Cidec*-F | This paper | Primers for qPCR | GTGTCCACTTGTGCCGTCTT |
| Sequence-based reagent | Mouse *Cidec*-R | This paper | Primers for qPCR | CTCGCTTGGTTGTCTTGATT |
| Sequence-based reagent | Mouse *Scd1*-F | This paper | Primers for qPCR | AGCTCTACACCTGCCTCTTCG |
| Sequence-based reagent | Mouse *Scd1*-R | This paper | Primers for qPCR | AGCCGTGCCTTGTAAGTTCTG |
| Sequence-based reagent | Mouse *Scd2*-F | This paper | Primers for qPCR | TACGGATATCGCCCCTACGA |
| Sequence-based reagent | Mouse *Scd2*-R | This paper | Primers for qPCR | GGAACTGCAAGACCCCACAC |
| Sequence-based reagent | Mouse *Col1a1*-F | This paper | Primers for qPCR | TTCAGCTTTGTGGACCTCCG |
| Sequence-based reagent | Mouse *Col1a1*-R | This paper | Primers for qPCR | GGACCCTTAGGCCATTGTGT |
| Sequence-based reagent | Mouse *Il-1β*-F | This paper | Primers for qPCR | ACAACTGCACTACAGGCTCC |
| Sequence-based reagent | Mouse *Il-1β*-R | This paper | Primers for qPCR | TGGGTGTGCCGTCTTTCATT |
| Sequence-based reagent | Mouse *Tnf-α*-F | This paper | Primers for qPCR | CGTCAGCCGATTTGCTATCT |
| Sequence-based reagent | Mouse *Tnf-α*-R | This paper | Primers for qPCR | CGGACTCCGCAAAGTCTAAG |
| Sequence-based reagent | Mouse *Tgf-β1*-F | This paper | Primers for qPCR | CCTCGAGACAGGCCATTTGT |
| Sequence-based reagent | Mouse *Tgf-β1*-R | This paper | Primers for qPCR | AAGGCCAGCTGACTGCTTT |
| Sequence-based reagent | Mouse *Il-6*-F | This paper | Primers for qPCR | AGTTGCCTTCTTGGGACTGA |
| Sequence-based reagent | Mouse *Il-6*-R | This paper | Primers for qPCR | TCCACGATTTCCCAGAGAAC |
| Sequence-based reagent | Mouse *SnoRNA U17*-F | This paper | Primers for qPCR | GTCCCTTTCCACAACGTTG |
| Sequence-based reagent | Mouse *SnoRNA U17*-R | This paper | Primers for qPCR | TTTCCTGCATGGTTTGTCTCC |
| Commercial assay or kit | BCA protein assay kit | LABLEAD | Cat#B5000 | |
| Commercial assay or kit | Lipofectamine 3000 Transfection Kit | Invitrogen | Cat#L3000-015 | |
| Commercial assay or kit | Seamless Assembly Cloning Kit | Abclonal | Cat#RM20523 | |
| Commercial assay or kit | High-Capacity cDNA Reverse Transcription Kit | Applied Biosystems | Cat#4368813 | |
| Commercial assay or kit | TIANprep Mini Plasmid Kit | TIANGEN | Cat#DP103-03 | |
| Commercial assay or kit | Endofree Maxi Plasmid Kit | TIANGEN | Cat#DP117 | |
| Commercial assay or kit | HiPure Gel Pure DNA Mini Kit | Magen | Cat#D2111-02 | |
| Commercial assay or kit | Equalbit 1x dsDNA HS Assay Kit | Vazyme | Cat#EQ121-01 | |
| Commercial assay or kit | Hyperactive Universal CUT&Tag Assay Kit for Illumina | Vazyme | Cat# TD903-01 | |

*Appendix 1 Continued on next page*

*Appendix 1 Continued*

| Reagent type (species) or resource | Designation | Source or reference | Identifiers | Additional information |
|---|---|---|---|---|
| Commercial assay or kit | TruePrep Index Kit V2 for Illumina | Vazyme | Cat#TD202 | |
| Commercial assay or kit | Sonication ChIP Kit | Abclonal | Cat#RK20258 | |
| Commercial assay or kit | RNA Immunoprecipitation(RIP) Kit | BersinBio | Cat#Bes5101 | |
| Commercial assay or kit | High Fatty Sample Total Cholesterol (TC) Content Assay Kit | APPLYGEN | Cat#E1026-105 | |
| Commercial assay or kit | High Fatty Sample Triglyceride(TG) Content Assay Kit | APPLYGEN | Cat#E1025-105 | |
| Commercial assay or kit | Mouse Insulin ELISA Kit | JINGMEI BIOTECHNOLOGY | JM-02862M1 | |
| Chemical compound, drug | Complete Tablets EDTA-free, EASYpack | Roche | Cat#4693132001 | |
| Chemical compound, drug | PMSF | Beyotime Biotechnology | Cat#ST506 | (1mM) |
| Chemical compound, drug | Palmitic acid (PA) | Sigma-Aldrich | Cat#P5585 | (1mM) |
| Chemical compound, drug | BSA (Fatty Acid & IgG Free, BioPremium) | Beyotime Biotechnology | Cat#ST025 | |
| Chemical compound, drug | Trizol | Invitrogen | Cat#15596018 | |
| Chemical compound, drug | Insulin | Sigma-Aldrich | Cat#I-5500 | |
| Chemical compound, drug | MG132 | AbMole | Cat#M1902 | (10μM) |
| Chemical compound, drug | Direct PCR Lysis Reagent (Tail) | Viagen Biotech | Cat#102T | |
| Chemical compound, drug | Collagenase II | Sigma-Aldrich | Cat#C6885-1G | (>100CDU/mL) |
| Chemical compound, drug | Oil Red O | Sigma-Aldrich | Cat#O0625 | |
| Chemical compound, drug | Biotin RNA Labelling Mix (Biotin-U) | Roche | Cat#11685597910 | |
| Chemical compound, drug | Yeast tRNA | Invitrogen | Cat#15401–011 | (100μg/mL) |
| Chemical compound, drug | Ribonucleoside Vanadyl Complexes (RVC) | Beyotime Biotechnology | Cat#R0107 | (400μM) |
| Chemical compound, drug | Recombinant RNase Inhibitor (RRI) | Takara | Cat#2313A | (100U/mL) |
| Chemical compound, drug | CA-630 (NP40) | Sigma-Aldrich | Cat#I3021 | (0.5%) |
| Chemical compound, drug | PPARγ antagonist (T0070907) | AbMole | Cat#M3044 | Primary hepatocytes were treated with T0070907 (15μM) mice injected intraperitoneally with T0070907 (1mg/kg) for 5days per week for 2months |
| Software, algorithm | SPSS statistics v17.0 | IBM Corporation | | http://www.spss.com.hk/software/statistics/ |
| Software, algorithm | ImageJ | ImageJ | | https://imagej.nih.gov/ij/ |
| Software, algorithm | GraphPad Prism 8 | GraphPad Software | | https://www.graphpad.com/ |
| Other | Mouse high fat diet | Research Diet | Cat#D12492 | Contain 60% fat for inducing obesity mice |
| Other | Disposable Iv indwelling needle | BD | Cat#381312 | For mouse liver perfusion vector |

