## [Editor Report · eLife assessment]

This study provides **useful** evidence substantiating a role for long noncoding RNAs in liver metabolism and organismal physiology. Using murine knockout and knock-in models, the authors invoke a previously unidentified role for the lncRNA Snhg3 in fatty liver. The revised manuscript has improved and most studies are backed by **solid** evidence but the study was found to be **incomplete** and will require future studies to substantiate some of the claims.

---

## [Referee Report · Reviewer #1 (Public Review)]

Summary:

In this manuscript the authors investigate the contributions of the long noncoding RNA snhg3 in liver metabolism and MAFLD. The authors conclude that liver-specific loss or overexpression of Snhg3 impacts hepatic lipid content and obesity through epigenetic mechanisms. More specifically, the authors invoke that nuclear activity of Snhg3 aggravates hepatic steatosis by altering the balance of activating and repressive chromatin marks at the Pparg gene locus. This regulatory circuit is dependent on a transcriptional regulator SNG1.

Strengths:

The authors developed a tissue specific lncRNA knockout and KI models. This effort is certainly appreciated as few lncRNA knockouts have been generated in the context of metabolism. Furthermore, lncRNA effects can be compensated in a whole organism or show subtle effects in acute versus chronic perturbation, rendering the focus on in vivo function important and highly relevant. In addition, Snhg3 was identified through a screening strategy and as a general rule the authors the authors attempt to follow unbiased approaches to decipher the mechanisms of Snhg3.

---

## [Referee Report · Reviewer #2 (Public Review)]

Through RNA analysis, Xie et al found LncRNA Snhg3 was one of the most down-regulated Snhgs by high fat diet (HFD) in mouse liver. Consequently, the authors sought to examine the mechanism through which Snhg3 is involved in the progression of metabolic dysfunction-associated fatty liver diseases (MASLD) in HFD-induced obese (DIO) mice. Interestingly, liver-specific Sngh3 knockout reduced, while Sngh3 over-expression potentiated fatty liver in mice on a HFD. Using the RNA pull-down approach, the authors identified SND1 as a potential Sngh3 interacting protein. SND1 is a component of the RNA-induced silencing complex (RISC). The authors found that Sngh3 increased SND1 ubiquitination to enhance SND1 protein stability, which then reduced the level of repressive chromatin H3K27me3 on PPARg promoter. The upregulation of PPARg, a lipogenic transcription factor, thus contributed to hepatic fat accumulation.

The authors propose a signaling cascade that explains how LncRNA sngh3 may promote hepatic steatosis. Multiple molecular approaches have been employed to identify molecular targets of the proposed mechanism, which is a strength of the study.

---

## [Author Response]

The following is the authors’ response to the previous reviews.

**Public Reviews:**

**Reviewer #1 (Public Review):**
Summary:In this manuscript the authors investigate the contributions of the long noncoding RNA snhg3 in liver metabolism and MAFLD. The authors conclude that liver-specific loss or overexpression of Snhg3 impacts hepatic lipid content and obesity through epigenetic mechanisms. More specifically, the authors invoke that nuclear activity of Snhg3 aggravates hepatic steatosis by altering the balance of activating and repressive chromatin marks at the Pparg gene locus. This regulatory circuit is dependent on a transcriptional regulator SNG1.Strengths:The authors developed a tissue specific lncRNA knockout and KI models. This effort is certainly appreciated as few lncRNA knockouts have been generated in the context of metabolism. Furthermore, lncRNA effects can be compensated in a whole organism or show subtle effects in acute versus chronic perturbation, rendering the focus on in vivo function important and highly relevant. In addition, Snhg3 was identified through a screening strategy and as a general rule the authors the authors attempt to follow unbiased approaches to decipher the mechanisms of Snhg3.Weaknesses:Despite efforts at generating a liver-specific knockout, the phenotypic characterization is not focused on the key readouts. Notably missing are rigorous lipid flux studies and targeted gene expression/protein measurement that would underpin why loss of Snhg3 protects from lipid accumulation. Along those lines, claims linking the Snhg3 to MAFLD would be better supported with careful interrogation of markers of fibrosis and advanced liver disease. In other areas, significance is limited since the presented data is either not clear or rigorous enough. Finally, there is an important conceptual limitation to the work since PPARG is not established to play a major role in the liver.

We thank the reviewer for the nice comment. As the reviewer comment, the manuscript still exists some shortcomings, we added partial shortcomings in the section of Discussion, please check them in the third paragraph on p17 and the first paragraph on p18.

We agree the reviewer comment, there are still conflicting conclusions about the role of PPARγ in MASLD. We had discussed it in the section of Discussion, please check them in the first paragraph on p13.

**Reviewer #2 (Public Review):**
Through RNA analysis, Xie et al found LncRNA Snhg3 was one of the most down-regulated Snhgs by high fat diet (HFD) in mouse liver. Consequently, the authors sought to examine the mechanism through which Snhg3 is involved in the progression of metabolic dysfunction-associated fatty liver diseases (MASLD) in HFD-induced obese (DIO) mice. Interestingly, liver-specific Sngh3 knockout reduced, while Sngh3 over-expression potentiated fatty liver in mice on a HFD. Using the RNA pull-down approach, the authors identified SND1 as a potential Sngh3 interacting protein. SND1 is a component of the RNA-induced silencing complex (RISC). The authors found that Sngh3 increased SND1 ubiquitination to enhance SND1 protein stability, which then reduced the level of repressive chromatin H3K27me3 on PPARg promoter. The upregulation of PPARg, a lipogenic transcription factor, thus contributed to hepatic fat accumulation.The authors propose a signaling cascade that explains how LncRNA sngh3 may promote hepatic steatosis. Multiple molecular approaches have been employed to identify molecular targets of the proposed mechanism, which is a strength of the study. There are, however, several potential issues to consider before jumping to the conclusion.(1) First of all, it's important to ensure the robustness and rigor of each study. The manuscript was not carefully put together. The image qualities for several figures were poor, making it difficult for the readers to evaluate the results with confidence. The biological replicates and numbers of experimental repeats for cell-based assays were not described. When possible, the entire immunoblot imaging used for quantification should be presented (rather than showing n=1 representative). There were multiple mis-labels in figure panels or figure legends (e.g., Fig. 2I, Fig. 2K and Fig. 3K). The b-actin immunoblot image was reused in Fig. 4J, Fig. 5G and Fig. 7B with different exposure times. These might be from the same cohort of mice. If the immunoblots were run at different times, the loading control should be included on the same blot as well.

We thank the reviewer for the detailed comment. We have provided the clear figures in revised manuscript, please check them.

The biological replicates and numbers of experimental repeats for cell-based assays had been updated and please check them in the manuscript.

The entire immunoblot imaging used for quantification had been provided in the primary data. Please check them.

The original Figure 2I, Figure 2K, Figure 3K have been revised and replaced with new Figure 2F, 2H, 3H, and their corresponding figure legends has also been corrected in revised manuscript.

The protein levels of CD36, PPARγ and β-ACTIN were examined at the same time and we had revised the manuscript, please check them in revised Figure 7B and C.

(2) The authors can do a better job in explaining the logic for how they came up with the potential function of each component of the signaling cascade. Sngh3 is down-regulated by HFD. However, the evidence presented indicates its involvement in promoting steatosis. In Fig. 1C, one would expect PPARg expression to be up-regulated (when Sngh3 was down-regulated). If so, the physiological observation conflicts with the proposed mechanism. In addition, SND1 is known to regulate RNA/miRNA processing. How do the authors rule out this potential mechanism? How about the hosting snoRNA, Snord17? Does it involve in the progression of NASLD?

We thank the reviewer for the detailed comment. In this study, although the expression of *Snhg3* was decreased in DIO mice, *Snhg3* deficiency decreased the expression of hepatic PPARγ and alleviated hepatic steatosis in DIO mice, and *Snhg3* overexpression induced the opposite effect, which led us to speculate that the downregulation of *Snhg3* in DIO mice might be a stress protective reaction to high nutritional state, but the specific details need to be clarified. This is probably similar to FGF21 and GDF15, whose endogenous expression and circulating levels are elevated in obese humans and mice despite their beneficial effects on obesity and related metabolic complications (Keipert and Ost, 2021). We had added the content in the Discussion section, please check it in the second paragraph on p12.

SND1 has multiple roles through associating with different types of RNA molecules, including mRNA, miRNA, circRNA, dsRNA and lncRNA. We agree with the reviewer good suggestion, the potential mechanism of SND1/lncRNA-*Snhg3* involved in hepatic lipid metabolism needs to be further investigated. We also discussed the limitation in the manuscript and please refer the section of Discussion in the third paragraph on p17.

*Snhg3* serves as host gene for producing intronic U17 snoRNAs, the H/ACA snoRNA. A previous study found that cholesterol trafficking phenotype was not due to reduced *Snhg3* expression, but rather to haploinsufficiency of U17 snoRNA (Jinn et al., 2015). Additionally, knockdown of U17 snoRNA *in vivo* protected against hepatic steatosis and lipid-induced oxidative stress and inflammation (Sletten et al., 2021). In this study, the expression of U17 snoRNA decreased in the liver of DIO *Snhg3*-HKO mice and remain unchanged in the liver of DIO *Snhg3*-HKI mice, but overexpression of U17 snoRNA had no effect on the expression of SND1 and PPARγ (figure supplement 5A-C), indicating that *Sngh3* induced hepatic steatosis was independent on U17 snoRNA. We had discussed it in revised manuscript, please refer to p15 of the Discussion section.

References

JINN, S., BRANDIS, K. A., REN, A., CHACKO, A., DUDLEY-RUCKER, N., GALE, S. E., SIDHU, R., FUJIWARA, H., JIANG, H., OLSEN, B. N., SCHAFFER, J. E. & ORY, D. S. 2015. snoRNA U17 regulates cellular cholesterol trafficking. *Cell Metab,* 21**,** 855-67. DIO:10.1016/j.cmet.2015.04.010, PMID:25980348

KEIPERT, S. & OST, M. 2021. Stress-induced FGF21 and GDF15 in obesity and obesity resistance. *Trends Endocrinol Metab,* 32**,** 904-915. DIO:10.1016/j.tem.2021.08.008, PMID:34526227

SLETTEN, A. C., DAVIDSON, J. W., YAGABASAN, B., MOORES, S., SCHWAIGER-HABER, M., FUJIWARA, H., GALE, S., JIANG, X., SIDHU, R., GELMAN, S. J., ZHAO, S., PATTI, G. J., ORY, D. S. & SCHAFFER, J. E. 2021. Loss of SNORA73 reprograms cellular metabolism and protects against steatohepatitis. *Nat Commun,* 12**,** 5214. DIO:10.1038/s41467-021-25457-y, PMID:34471131

(3) The role of PPARg in fatty liver diseases might be a rodent-specific phenomenon. PPARg agonist treatment in humans may actually reduce ectopic fat deposition by increasing fat storage in adipose tissues. The relevance of the finding to human diseases should be discussed.

We thank the reviewer for the detailed comment. We agree the reviewer comment, there are still conflicting conclusions about the role of PPARγ in MASLD. We had discussed it in the section of Discussion, please check them in the first paragraph on p13.

**Recommendations for the authors:**

**Reviewer #1 (Recommendations For The Authors):**
I do not have further recommendations beyond what I mentioned in the original review. The authors have not adequately addressed all the issues but the manuscript has improved and the overall strength of evidence is now solid from incomplete.

We appreciate positive feedback from the reviewer. While we acknowledge that the updated manuscript has significantly improved, we recognize that it remains incomplete and additional details regarding *Snhg3* will be warranted in our future studies. Moreover, we have discussed those potential weakness in the section of Discussion (please refer in the third paragraph on p17 and the first paragraph on p18).

**Reviewer #2 (Recommendations For The Authors):**
The authors have provided explanations and some new data to clarify the comments from the first submission. They have also included the original immunoblots for all the experimental repeats. The CHX protein stability results shown in Fig. 5J were not consistent between experiments, perhaps because the difference was subtle. The results on PPARg protein expression were not clearcut. The inclusion of a PPARg knockdown control would be helpful to validate the specificity of the antibody. Of note, the immunoblots used for Fig. 5I (PA treated) repeats 2, 4 and 1 were identical to those of Fig. 7F repeats 3, 1 and 5. The authors should provide an explanation for the potential issue.

We thank the further comments and suggestions from the reviewer. We agree with the reviewer comment about *Snhg3*-mediated SND1 protein stability. In this study, *Snhg3* promoted the protein, not mRNA, level of SND1, but *Snhg3* subtly increased the SND1 protein stability. We revised the description in the manuscript, “Meanwhile, *Snhg3* regulated the protein, not mRNA, expression of SND1 in vivo and in vitro by mildly promoting the stability of SND1 protein (Figures 5G-I).” This revision can be found in the second paragraph on p9. While our findings indicated that *Snhg3* can influence SND1 expression at the protein level, we acknowledge the possibility of additional mechanisms contributing to this complex regulatory network. Therefore, further investigation is necessary to clarify whether Snhg3 regulates SND1 protein expression through other potential mechanisms. In light of this, we have added it in the Discussion section. Please refer to the second paragraph on p16.

In this study, the protein level of PPARγ (molecular weight ~57 kDa) was detected using anti-PPARγ antibody (Abclonal, Cat. A11183), which has been used to determine PPARγ protein expression in 13 published papers as showed in the ABclonal Technology Co., Ltd. (https://abclonal.com.cn/catalog/A11183). And the specificity of this antibody has been validated in Zhang’s study by PPARγ knockdown (Zhang et al., 2019). In our study, hepatic PPARγ protein sometimes showed two bands (~ 57kDa and > 75kDa) using this antibody. It is well established that the PPARγ gene encodes two protein isoforms (PPARγ1, a 477 amino acid protein, and PPARγ2, a 505 amino acid protein) via differential promoter usage and alternative splicing (Gene: Pparg (ENSMUSG00000000440) - Transcript comparison - Mus_musculus - Ensembl genome browser 112) (Hernandez-Quiles et al., 2021). The molecular weight difference between PPARγ1 and PPARγ2 is about 3kd. Therefore, we consider that the band shown larger than 75kd in our study is likely nonspecific. In line with the reviewer’s suggestion, the antibody’s specificity could be further validated by knockdown or knockout of PPARγ in the future.

We thank the reviewer for the detailed comment. In this study, we tested the effect of *Snhg3* overexpression on SND1 protein level and the effect of *Snhg3* or *Snd1* overexpression on PPARγ protein level in Hepa1-6 cells by transfecting with *Snhg3*, SND1 and the control, respectively. The results indicated that overexpression of *Snhg3* promoted the protein levels of SND1 and PPARγ, and overexpression of SND1 also induced the protein level of PPARγ. Considering scholarly and professional thinking and writing, we firstly showed that overexpression of *Snhg3* promoted the protein level of SND1 in *Figure 5I*, followed by demonstrating that the overexpression of *Snhg3* or SND1 elicited PPARγ expression in Figures 7F. However, we acknowledge that this order of presentation may cause confusion. In fact, these experiments were repeatedly performed by multiple times, and we have provided the new original western blot data and analysis for Figure 5I (PA treatment) for further clarification. Please check them.

References

HERNANDEZ-QUILES, M., BROEKEMA, M. F. & KALKHOVEN, E. 2021. PPARgamma in Metabolism, Immunity, and Cancer: Unified and Diverse Mechanisms of Action. *Front Endocrinol (Lausanne),* 12, 624112. DIO:10.3389/fendo.2021.624112, PMID:33716977

ZHANG, Z., ZHAO, G., LIU, L., HE, J., DARWAZEH, R., LIU, H., CHEN, H., ZHOU, C., GUO, Z. & SUN, X. 2019. Bexarotene Exerts Protective Effects Through Modulation of the Cerebral Vascular Smooth Muscle Cell Phenotypic Transformation by Regulating PPARgamma/FLAP/LTB(4) After Subarachnoid Hemorrhage in Rats. *Cell Transplant,* 28, 1161-1172. DIO:10.1177/0963689719842161, PMID:31010302